# RECITATION-AUGMENTED LANGUAGE MODELS

**Zhiqing Sun**[1,2*]**, Xuezhi Wang**[1]**, Yi Tay**[1]**, Yiming Yang**[2]**, Denny Zhou**[1]
[1]Google Research, Brain Team
[2]Language Technologies Institute, Carnegie Mellon University

## ABSTRACT

We propose a new paradigm to help Large Language Models (LLMs) generate more accurate factual knowledge without retrieving from an external corpus, called RECITation-augmented gEneration (RECITE). Different from retrieval-augmented language models that retrieve relevant documents before generating the outputs, given an input, RECITE first recites one or several relevant passages from LLMs' own memory via sampling, and then produces the final answers. We show that RECITE is a powerful paradigm for knowledge-intensive NLP tasks. Specifically, we show that by utilizing recitation as the intermediate step, a recite-and-answer scheme can achieve new state-of-the-art performance in various closed-book question answering (CBQA) tasks. In experiments, we verify the effectiveness of RECITE on four pre-trained models (PaLM, UL2, OPT, and Codex) and three CBQA tasks (Natural Questions, TriviaQA, and HotpotQA). Our code is available at https://github.com/Edward-Sun/RECITE.

## 1 INTRODUCTION

Large language models (LLMs) have achieved impressive in-context few-shot performance on knowledge-intensive NLP tasks (Brown et al., 2020; Rae et al., 2021; Hoffmann et al., 2022; Chowdhery et al., 2022). For example, in open-domain question answering (Chen et al., 2017), demonstrated by only a few examples of question-answer pairs, LLMs are able to answer arbitrary factoid questions (Joshi et al., 2017; Yang et al., 2018; Kwiatkowski et al., 2019). Recent research (Guu et al., 2020; Lewis et al., 2020; Izacard et al., 2022) shows that retrieval-augmentation can further improve LLMs' performance on knowledge-intensive tasks by conditioning the LLMs on retrieved relevant passages from an external corpus.

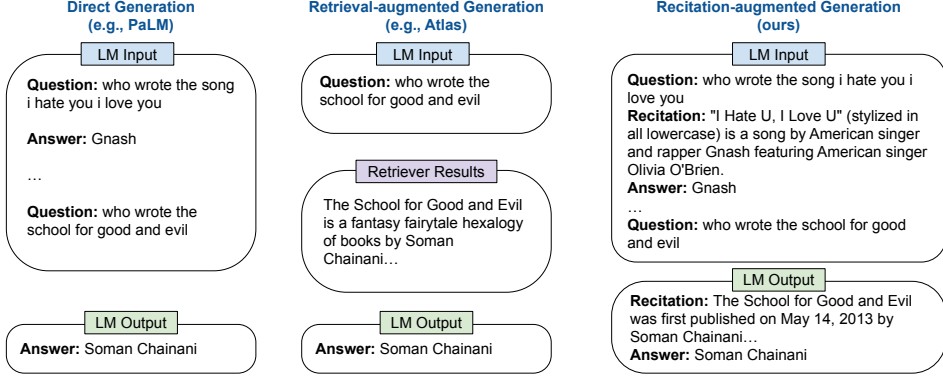

Figure 1: Illustration of evaluating (few-shot) open-domain question answering with (closed-book) direct generation (Chowdhery et al., 2022), (open-book) retrieval-augmented generation (Izacard et al., 2022), and (closed-book) recitation-augmented generation (ours).

---

* Work done during internship at Google.

This paper proposes a new paradigm to help LLMs generate more accurate factual knowledge without retrieving from an external corpus, called RECITation-augmented gEneration (RECITE), wherein we tackle knowledge-intensive NLP tasks by first reciting relevant information and then generating the outputs. Such a two-step paradigm decomposes the original knowledge-intensive task into two sub-tasks: knowledge-recitation and task-execution, where the former can be regarded as a form of intermediate knowledge retrieval step (from the model weights), while the latter is the execution step that produces the final outputs.

The motivation of introducing an additional knowledge-recitation step comes from our observation that while few-shot prompting can help LLMs execute specific NLP tasks, these tasks are usually not in a similar form as the original causal language modeling pre-training objective. This hinders LLMs from effectively reciting knowledge from their memory (Carlini et al., 2021). Consider a student taking a closed-book exam that contains knowledge-intensive questions, for example, **"what is the tenth decimal of $\pi$?"**. They typically cannot directly answer this question because in studying stage (in analogy to the language modeling pre-training stage for LLMs), it is highly unlikely that they would read "the tenth decimal of $\pi$ is 5". However, there can be some sentences like "the first $N$ digits of $\pi$ are 3.14159 26535..." existing in the textbook that can be recited by the student. Therefore, a student can possibly answer this question in a recite-and-answer scheme: **"The first 10 digits of $\pi$ are 3.14159 26535. So the answer is 5"**. Here, the knowledge-recitation step can serve as an intermediate step that mimics the language modeling pre-training task, and thus better helps the LLM to generate factual knowledge.

We verify the effectiveness of our recitation-augmented generation on few-shot Closed-Book Question Answering (CBQA) tasks (referred as **recite-and-answer** in the CBQA context), as illustrated in Figure 1. CBQA is an attractive open-domain QA task in that a fully parameterized LM can generate answers directly without an external corpus or separate retrieval models (Roberts et al., 2020). We show that the proposed recite-and-answer scheme is an effective method for CBQA and compatible with other techniques for boosting few-shot performance of LLMs. We also show that, in addition to improving the few-shot in-context learning performance of RECITE-enhanced LLM, fine-tuning the pre-trained LLMs on synthetic generated question-passage pairs can further improve the recitation performance and lead to a better downstream QA accuracy.

Experiments on four large language models (PaLM (Chowdhery et al., 2022), UL2 (Tay et al., 2022a), OPT (Zhang et al., 2022)), and Codex (Chen et al., 2021) show that a recite-and-answer scheme can improve performance on various types of CBQA tasks, including Wikipedia-based single-hop QA (Natural Questions, Kwiatkowski et al. 2019), trivia questions (TriviaQA, Joshi et al. 2017), and Wikipedia-based multi-hop QA (HotpotQA, Yang et al. 2018).

## 2 RELATED WORK

### 2.1 OPEN-DOMAIN QUESTION ANSWERING

Open-domain question answering (Prager et al., 2007) refers to the task of generating answers for arbitrary context-free questions. In the open-book setting, it is typically assumed that the QA model can find the answer in an external corpus, e.g., Wikipedia (Chen et al., 2017; Izacard & Grave, 2021) or web pages (Lazaridou et al., 2022). This is in analogy as taking an open-book exam where students can search over an external knowledge corpus. The standard pipeline (Chen et al., 2017; Izacard & Grave, 2021; 2020) usually consists of a learnable or non-learnable document retriever module and a learnable neural network-based reader module.

In the closed-book setting, the QA model is not allowed to access any external knowledge, and needs to store all the knowledge in its parameters. It has been recently observed that large-scale pre-trained language models (Devlin et al., 2019; Radford et al., a; Yang et al., 2019b) can internalize a sort of implicit "knowledge base" after pre-training (Petroni et al., 2019; Jiang et al., 2020; Talmor et al., 2020). Roberts et al. (2020) show that after fine-tuning on open-book question-answer pairs, T5 (Raffel et al., 2020) can answer a large portion of knowledge-intensive questions. This is similar as taking a closed-book exam. However, Lewis et al. (2021) found that the high performance is mainly due to training set question memorization. Wang et al. (2021) also found that it is still challenging for relatively small-scale pre-trained language models like RoBERTa (Liu et al., 2019) or GPT-2 (Radford et al., b) to answer closed-book questions.

In this work, we focus on evaluating the CBQA performance of large language models (LLMs) in the few-shot setting, which ideally minimizes the bias of train-test overlapping (Liu et al., 2021). We propose a recite-and-answer scheme, which is similar to a student first recite the factoid knowledge about the question, and then answer the question.

## 2.2 IN-CONTEXT FEW-SHOT LEARNING

Large language models (LLMs) such as GPT-3 (Brown et al., 2020) have the surprising ability to do in-context learning, where the model learns to do new tasks simply by being prompted a few exemplars. The LLMs learn from these exemplars without being explicitly pre-trained for in-context learning and without any gradient updates or fine-tuning. Recent study showed that such ability improves with the scaling of both model size (Brown et al., 2020; Rae et al., 2021; Chowdhery et al., 2022) and number of tokens for training (Hoffmann et al., 2022). When evaluated on knowledge-intensive question answering tasks, these models are usually evaluated in the closed-book setting, where the factoid knowledge are completely stored in the model parameters of dense LLMs.

Recently, Atlas (Izacard et al., 2022) shows that for knowledge-intensive NLP tasks, a relatively lite model with retrieval augmentations can achieve similar or even better performance through few-shot fine-tuning, which proves that memorization can be decoupled from generalization in LLMs. In contrast, we show that still a large underestimated amount of knowledge can be retrieved from LLMs' model weights through better-designed prompting.

## 2.3 RATIONALE-AUGMENTED REASONING

Ling et al. (2017) pioneer the work of solving math word problems by generating step-by-step human-readable solutions described by natural language and math equations before the final answer. That is fundamentally different from other works which directly generate the final answers or use formal languages. e.g. equations only, to illustrate the intermediate solving steps (Roy et al., 2016; Amini et al., 2019; Chen et al., 2019). Cobbe et al. (2021) extend (Ling et al., 2017) by constructing a much larger dataset to finetune a pre-trained large language model to solve math word problems and a parameterized ranker is trained to rank candidate solutions to improve the solving rate. Wei et al. (2022) propose chain-of-thought prompting which combines the idea of natural language rationales (Ling et al., 2017; Cobbe et al., 2021) with few-shot prompting (Brown et al., 2020).

In this work, instead of generating a chain of thought for multi-step reasoning questions, we decompose the process of answering a knowledge-intensive question into two steps: recite the relevant knowledge stored in the model parameters, and then answer the question.

## 2.4 MEMORIZATION IN LARGE LANGUAGE MODELS

Recent study shows that large language models can memorize its training data, and generate texts from training data given certain prompts (Carlini et al., 2021; 2022; Zhang et al., 2021; Kharitonov et al., 2021; Thakkar et al., 2020; Carlini et al., 2019; Tirumala et al., 2022). Most related to our work, Carlini et al. (2022) found that the memorization ability of LLMs significantly grows as the model capacity increases, the number of times an example has been duplicated, and the number of tokens of context used to prompt the model. While these works mainly analyze the fundamental properties of memorization in the exact setting, where exactly $N$ tokens are used as the prompt to reproduce the suffix of the prompt, our work relies on "fuzzy memorizaiton", where the prompts tend to not be exactly the same as the training data, but still improve the memorization accuracy.

The proposed recitation-augmented generation idea is also related to the line of work on utilizing Transformer memory as an information retrieval model (Tay et al., 2022b) and self-talk models for commonsense reasoning (Shwartz et al., 2020; Liu et al., 2022). Zhuang et al. (2022); Wang et al. (2022c); Zhou et al. (2022) proposed to augment documents at indexing time with a number of generated queries. Bevilacqua et al. (2022) proposed to directly generate n-grams grounded in one or multiple documents with constrained decoding.

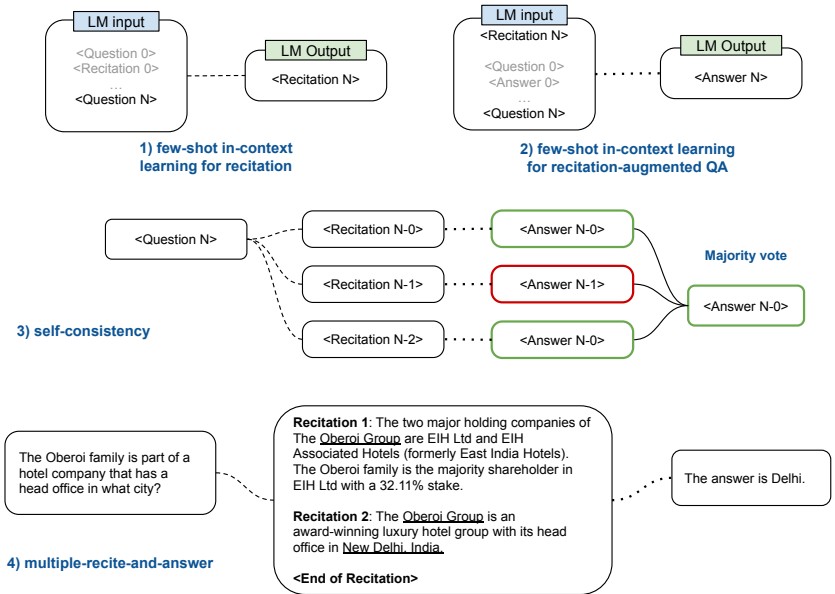

Figure 2: Illustration of prompt-based in-context learning for recitation generation, recitation-augmented question answering, self-consistency ensembling, and multiple-recite-and-answer for multi-hop questions (Sec. 3.1). In multiple-recite-and-answer scheme, the latter recitaiton can utilize the information from the previous ones, such as "Oberoi Group" in this case. The prompts for self-consistency and multi-hop recite-and-answer are dropped for brevity.

## 3 LEARNING TO RECITE FOR CLOSED-BOOK QUESTION ANSWERING

The goal of this paper is to mimic a human's ability to recite relevant factoid knowledge (McDaniel et al., 2009) before answering knowledge-intensive questions, such that these questions can be answered more accurately. In the following we describe our recite-and-answer scheme for few-shot closed-book question answering (CBQA), which consists of two components: (1) a evidence-recitation module for reciting relevant passages, and (2) a question-answering module for generating answers given the recited evidence. Notice that in this paper, we focus on few-shot setting, in which we assume only a few question-answer demonstrations are provided. In Natural Questions (Kwiatkowski et al., 2019) benchmark, since the questions are from queries issued to the Google search engine by multiple users, and thus can be regarded as unannotated data, we further assume that we have top-retrieved Wikipedia pages for these questions. The paragraphs in these top-retrieved Wikipedia pages will be used to generate synthetic paired question-recitation data for fine-tuning the LM (described in Section 3.2).

### 3.1 PROMPT-BASED RECITE-AND-ANSWER FOR QUESTION-ANSWERING

**Recitation-augmented question answering** We start with single-hop question answering (Kwiatkowski et al., 2019; Joshi et al., 2017), where the answers are usually supported by a specific document in the corpus, which is sometimes referred as evidence (Joshi et al., 2017). Different from chain-of-thought prompting (Wei et al., 2022) where a rationale is directly generated to explain the generated answer (Joshi et al., 2017; Narang et al., 2020; Lampinen et al., 2022), we propose to first recite a passage about the question, and then answer the question based on the recitation.

We propose a prompt-based learning-to-recite scheme by leveraging the LLM's in-context learning ability (Brown et al., 2020). We prompt the LLM with paired exemplars of questions and recited evidences, and the LLM can learn in an in-context manner to generate a recitation for an arbitrary question. To perform recitation-conditioned few-shot question answering, we append the recited passages at the beginning of the original question-answer exemplars as a single prompt, and then generate the final answer (Step 1 & 2 in Figure 2).

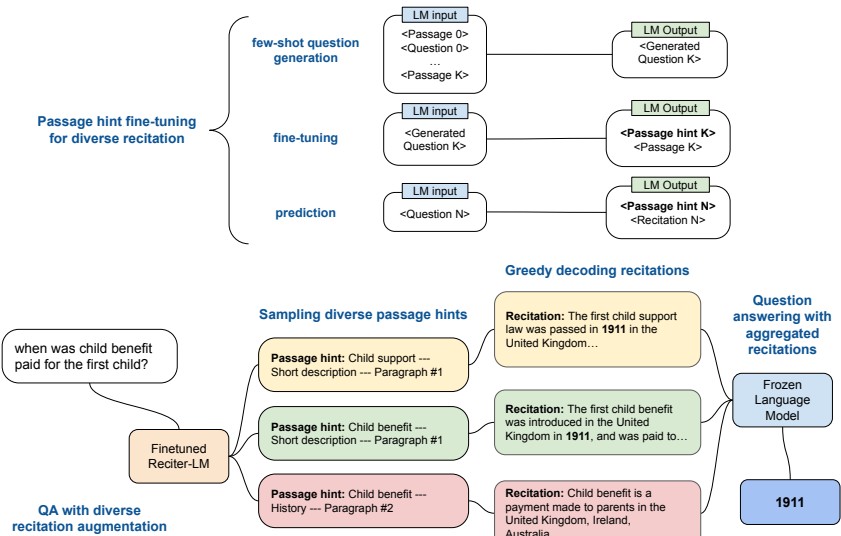

Figure 3: Illustration of question answering with diverse recitation and the corresponding few-shot question generation and fine-tuning processes.

**Self-consistency ensemble**  The factual knowledge about a question can appear in several places in the language model's training corpora. For example, the fact of "Queen Elizabeth II opened the London Bridge on 17 March 1973" can appear in both Wikipedia page "London Bridge" and page "March 1973", so it is highly likely that there exists knowledge from different articles that could lead to the same, correct answer. With this motivation, we argue that similar to multi-step reasoning in chain-of-thought, recitation-augmented question answering can also benefit from the self-consistency technique with multiple-path decoding (Wang et al., 2022b). Specifically, given an arbitrary question, we first use top-$k$ sampling to independently generate a few recitations, and then greedy decode the answer of the question based on the sampled recitations. Finally, we determine the optimal answer by taking a plurality/majority vote (Step 3 in Figure 2).

**Multiple-recite-and-answer for multi-hop question-answering**  Multi-hop question answering requires the QA system to find and reason over multiple supporting documents. However, the nature of recitation restricts us to recite passages from one article at a time. In order to apply the recite-and-answer scheme to solve multi-hop questions, we introduce multiple-recite-and-answer scheme (Step 4 in Figure 2), that is, given the multiple-hop question, we use the prompt words such as "Recitation 1" and "Recitation 2" to elicit the LLM to generate recitation passages on different topics. Since the multiple recited passages are generated in one-pass from the LLM decoding sequentially, the generation of later passages can effectively utilize the information both in the original question and the previous recited ones. Our multiple-recite-and-answer scheme for multi-hop question-answering is also compatible with the self-consistency technique, by applying top-$k$ sampling when generating multiple recitations and performing majority voting for the final answers.

## 3.2 PASSAGE HINT-BASED DIVERSIFIED RECITATION WITH FINE-TUNING

**Passage hint-based diversified recitation**  While the sampling-based recitation and self-consistency improves the robustness of recite-and-answer method, one argument for its inefficiency is that if the evidence-recitation module samples the wrong facts about the question, the question-answering module will not be able to figure it out and tend to generate the wrong answer. Therefore, on the one hand, we need to use a low sampling temperature to avoid generating recitations with wrong facts, on the other hand, we want to make sure the sampled recitations have enough diversity.

To tackle such a dilemma, we propose *passage hint-based diversified recitation*. We observe that in well-formed text knowledge bases, such as Wikipedia, we can usually find a unique passage hint for each passage, by concatenating the section titles and the in-section order of each passage. For example, the passage hint of the second passage in Section 5.2 "Enforcement" of Wikipedia page

"Child support" would be "Child support — Compliance and enforcement issues — Enforcement — Paragraph #2". In passage hint-based diversified recitation, we first use *sampling* to generate a diverse set of passage hints, and then use *greedy decoding* to ensure the factual accuracy of the contents in each passage.

Since each passage hint corresponds to a unique passage, we can first de-duplicate the passage hints and then generate the full passages to get more diverse recitation passages. Furthermore, as the recited passages are less likely to be similar due to unique passage hints, inspired by recent progress on question-answering with multiple retrieved passages (Izacard & Grave, 2021), we use aggregated diverse recitations as a single context, and generate the answer with a few more question-answer pair demonstrations. Figure 3 (lower) illustrates the recite-and-answer scheme with passage hint-based diversified recitation.

**Fine-tuning on few-shot generated questions**    We found that although the training data of many existing LLMs (Devlin et al., 2019; Chowdhery et al., 2022) contains the Wikipedia corpora, which are usually regarded as the factoid documents for knowledge-intensive question answering tasks (Joshi et al., 2017; Kwiatkowski et al., 2019), the section titles are usually not explicitly included in training. This makes the pre-trained LLM hard to discover the mapping from the question to the passage hint, and to the full passage merely by few-shot prompting.

To address this issue, we propose an additional fine-tuning stage to adapt LLMs to learn such mappings. Assuming we have access to not only a few question-answer pairs, but also the top-retrieved Wikipedia pages for queries issued to the Google search engine by multiple users (Kwiatkowski et al., 2019), we can use few-shot prompting to generated synthetic question-hint-passage pairs and then finetune the LLMs on the generated data.

Specifically, we use the ground-truth evidence and question pairs as the prompt, and generate new questions with in-context learning for randomly sampled passages from Wikipedia pages. Next, based on the few-shot generated questions, we train the LLM to predict the original passage hint, as well as the passage content. Figure 3 (upper) illustrates the whole process of passage hint fine-tuning.

## 4    EXPERIMENTS

In this section, we report empirical evaluations of our proposed RECITE with recite-and-answer schemes on a diverse set of few-shot closed-book question answering tasks and different language models with varying scales.

### 4.1    EXPERIMENTAL SETUP

#### 4.1.1    EVALUATION DATASETS

**Natural Questions**    Natural Questions (Kwiatkowski et al., 2019) consists of questions aggregated from the Google search engine and the answers from the Wikipedia page in the top 5 search results. We treat it as a single-hop question answering task. Since Natural Questions contains the so-called "long answer" annotations, which is a whole HTML bounding box containing enough information to infer the answer, we directly utilize the "long answer" as the ground-truth recitation exemplars in our prompt (Sec. 3.1). In order to make a direct comparison with recent LLMs (Chowdhery et al., 2022; Izacard et al., 2022), we evaluate our methods in 5-shot and 64-shot settings.

**TriviaQA**    TriviaQA dataset (Joshi et al., 2017) is constructed by collecting Trivia enthusiast authored question-answer pairs and their retrospectively collected evidence. Since there is no obvious way to collect a "long answer" in the retrospective evidence documents (the exact appearances of the answer may contain enough information to infer the answer), we evaluate TriviaQA in the single-hop 5-shot setting, and manually compose the recitation passage from Wikipedia for 5 randomly sampled training questions. The concrete prompt can be found in the appendix.

**HotpotQA**    HotpotQA (Yang et al., 2018) is designed to explicitly test QA systems' ability to perform multi-hop reasoning. It is collected by explicitly composing questions requiring reasoning

Table 1: Performance comparison on Natural Questions (NQ), TriviaQA, and HotpotQA. The number of shots for each task are mentioned in parenthesis.

| | | PaLM-62B EM / F1 | UL2-20B EM / F1 | OPT-30B EM / F1 | Codex-002 EM / F1 |
|---|---|---|---|---|---|
| NQ | Standard-prompting (direct) | 25.76 / 36.47$_{(5)}$
28.98 / 40.13$_{(64)}$ | 10.16 / 20.17$_{(5)}$
12.70 / 21.97$_{(16)}$ | 14.97 / 22.93$_{(5)}$ | 31.45 / 44.75$_{(5)}$ |
| | Recite-and-answer (20-path) | **28.70 / 39.76**$_{(5)}$
**31.34 / 42.48**$_{(64)}$ | **14.16 / 23.13**$_{(5)}$
**14.94 / 24.29**$_{(16)}$ | **17.84 / 26.74** $_{(5)}$ | **35.84 / 49.12**$_{(5)}$ |
| TriviaQA | Standard-prompting (direct) | 65.38 / 71.85$_{(5)}$ | 48.73 / 54.32$_{(5)}$ | 45.90 / 50.68$_{(5)}$ | 81.84 / 86.09$_{(5)}$ |
| | Recite-and-answer (20-path) | **65.84 / 72.10**$_{(5)}$ | **53.42 / 58.69**$_{(5)}$ | **49.02 / 54.22**$_{(5)}$ | **83.50 / 88.03**$_{(5)}$ |
| HotpotQA | Standard-prompting (direct) | 20.51 / 28.90$_{(4)}$ | 16.99 / 24.99$_{(4)}$ | 16.70 / 25.21$_{(4)}$ | 28.32 / 39.03$_{(4)}$ |
| | Chain-of-thought (20-path) | 23.73 / 32.80$_{(4)}$ | 17.68 / 24.87$_{(4)}$ | 16.89 / 24.03$_{(4)}$ | 34.38 / 45.50$_{(4)}$ |
| | Recite-and-answer (20-path) | **26.46 / 35.67**$_{(4)}$ | **19.04 / 27.32**$_{(4)}$ | **17.77 / 26.58**$_{(4)}$ | **37.11 / 48.37**$_{(4)}$ |

about multiple supporting context documents. Following Wang et al. (2022a), we evaluate HotpotQA as a multi-hop question answering task in the 4-shot setting. But instead of chain-of-thought prompting as in (Wang et al., 2022a), we use multiple-recite-and-answer (Sec. 3.1) to achieve multi-step reasoning. We also provide the concrete prompt in the appendix.

**Metrics** We calculate the Exact Matching (EM) and F1 scores for the normalized answers, while the specific text normalization applied on each dataset can be slightly different.

### 4.1.2 PRE-TRAINED LANGUAGE MODELS

We evaluate the effectiveness of RECITE on four langauge models: PaLM, UL2 (Tay et al., 2022a), OPT (Zhang et al., 2022), and Codex (Brown et al., 2020; Ouyang et al., 2022; Chen et al., 2021). Due to the space limit, the detailed descriptions of them are provided in Appendix D.

## 4.2 EXPERIMENTS

We use the test split for all tasks if the test split is available and has labels for evaluation, otherwise we use the dev split. In addition, TriviaQA and HotpotQA are too large to run large language models on, so we used the first 1,024 data points for evaluation.

### 4.2.1 PROMPT-BASED RESULTS

We report the single-hop closed-book question answering (CBQA) evaluation results on Natural Questions (NQ) and TriviaQA and the multi-hop CBQA evaluation results on HotpotQA. In Tab. 1, we report the results with prompt-based in-context learning and self-consistency.

From the tables, we can see that the proposed recite-and-answer scheme can significantly improve the CBQA performance on both datasets with various pre-trained language models. While the performance improvements on NQ is more consistent across different language models, we find that the improvements from recite-and-answer is more significant on smaller language models on TriviaQA. Our hypothesis is that the Trivia-style question usually contains more contextual information in the question, thus weakened the effectiveness of recitation for strong LLMs like PaLM.

Besides, we can see that the recite-and-answer scheme can outperform the chain-of-thought prompting performance on the multi-hop reasoning task. Interestingly, we also find that for LLMs that have large gains from chain-of-thought (i.e., PaLM), they also have large improvements from recite-and-answer.

### 4.2.2 RESULTS OF PASSAGE HINT-BASED DIVERSIFIED RECITATION

For Natural Questions dataset, since it has the collection of top-retrieved Wikipeida pages corresponding to the unannotated queries issued to the Google search engine, we additionally report the diversified recitation results of fine-tuned PaLM model in Tab. 2. From the table, we find that diversified recitation can further improve the performance of PaLM on the NQ dataset.

Table 2: Performance comparison of PaLM-62B on Natural Questions (NQ) dataset with standard-prompting, recite-and-answer with self-consistency sampling, and recite-and-answer with diversified recitation. The number of shots for each task are mentioned in parenthesis.

|  | EM / F1$_{(5)}$ | EM / F1$_{(64)}$ |
| --- | --- | --- |
| Standard-prompting (direct) | 25.76 / 36.47 | 28.98 / 40.13 |
| Recite-and-answer (20-path) | 28.70 / 39.76 | 31.34 / 42.48 |
| Recite-and-answer w/ diversified recitation (20-path) | **32.20 / 44.02** | **33.23 / 45.29** |

## 4.3 ANALYSIS

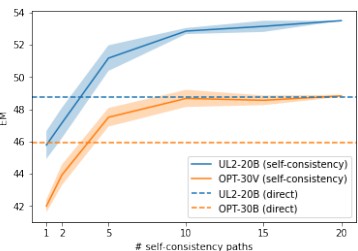 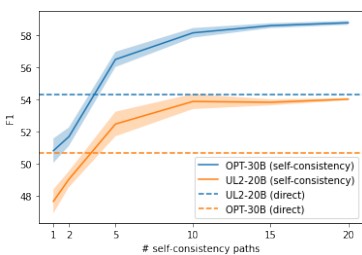

Figure 4: TriviaQA EM/F1 on OPT-30B and UL2-20B with different # of self-consistency paths.

### 4.3.1 ON THE NUMBER OF SELF-CONSISTENCY PATHS

We analyze how the number of passages recited would affect the performance of recite-and-answer under the self-consistency setting. Due to the costly inference of LLMs, we first sample up to $k = 20$ recitation passages, and then apply self-consistency to a randomly selected subset of recitations to simulate less paths. For each number of self-consistency paths, we evaluate the randomly selected subsets five times and report the mean and standard deviation. We conduct an analysis on OPT-30B and UL2-20B on the TriviaQA dataset and report the results in Fig. 4. We can see that sampling more recitation passages tends to improve the recite-and-answer performance, while less randomness is observed with more self-consistency paths.

### 4.3.2 ON THE ROBUSTNESS OF FEW-SHOT EXEMPLARS

A well-known problem of in-context few-shot learning is its instability to the choices of exemplars and their orders (Zhao et al., 2021). We evaluate the robustness of standard prompting and our recite-and-answer method with 5 random seeds and report the mean and standard deviation of UL2 model running on the TriviaQA dataset in Tab. 6. The 5-shot exemplars are randomly sampled and shuffled for each seed. From the table, we can see that with recitation sampling, recite-and-answer exhibits similar robustness (in terms of small performance deviation) as standard prompting under different random seeds and numbers of self-consistency paths. The overall gains by recite-and-answer are significant compared to standard prompting regardless of the choice of few-shot exemplars.

### 4.3.3 RECITATION V.S. RETRIEVAL V.S. GROUND-TRUTH

One may ask without the external corpus, whether the quality of recited passages with LLMs is better than simple retrieval models, e.g., BM25 (Robertson et al., 2009)[1]. To answer this question, we evaluate the few-shot question-answering performance of UL2 and Codex on three kinds of context passages: retrieval, recitation, and ground-truth. We report the results on first 1024 validation examples in Natural Questions (NQ) dataset, since it is the only dataset that contains the "long answer" annotation that can be regarded as ground-truth context passage. From Tab. 3, we can see that the classic retrieval model, i.e., BM25, is still a very strong baseline for collecting information from the corpus. Nonetheless, compared to BM25, our recite-and-answer still achieves a quite competitive performance via generation only and without using any external corpus. Besides, we find that

---

[1]We use the pre-indexed "enwiki-paragraphs" corpus in the pyserini package (https://github.com/castorini/pyserini), which is originally designed for BERTserini (Yang et al., 2019a).

Table 3: Natural Questions (NQ) results with different context passages.

|  | UL2-20B$_{(5)}$ EM / F1 | Codex-002$_{(5)}$ EM / F1 |
|---|---|---|
| No passage | 10.16 / 20.17 | 31.45 / 44.75 |
| Ground-truth passage | 41.02 / 55.73 | 49.32 / 64.32 |
| BM25-Retrieval (Top-1) | 16.31 / 27.66 | 33.20 / 47.45 |
| LM-Recitation$_{(5)}$ (20-path) | 14.16 / 23.13 | 35.84 / 49.12 |

Table 4: Per-question error analysis on TriviaQA.

|  | UL2-20B$_{(5)}$ | OPT-30B$_{(5)}$ |
|---|---|---|
| Hits@Majority | 53.42% | 49.02% |
| Not Recit. | 21.09% | 22.27% |
| Hits@20-Recit. | 5.66% | 8.01% |
| Hits@20-Path | 19.82% | 20.07% |

Table 5: Per-path error analysis on TriviaQA.

| Recit. | Ans. | UL2-20B$_{(5)}$ | OPT-30B$_{(5)}$ |
|---|---|---|---|
| ✓ | ✓ | 33.60% | 30.06% |
| ✓ | ✗ | 7.87% | 9.79% |
| ✗ | ✓ | 12.10% | 12.57% |
| ✗ | ✗ | 46.44% | 47.58% |

stronger models (i.e., Codex) tend to benefit more from the the model's own recitation than BM25 retrieved context.

### 4.3.4 ERROR ANALYSIS

We perform an error analysis on the 1024 evaluation examples in the TriviaQA dataset. We classify the errors into three categories: 1) Not Recit., i.e., the correct answer is not recited in any of the 20 recited passages in self-consistency. 2) Hits@20-Recit., i.e., the correct answer can be found in one of the recited passage, but does not appear in the QA module's outputs. 3) Hits@20-Path, i.e., the correct answer is one of the final outputs of the 20 self-consistency paths, but it does not have the majority votes. The correct final answer is marked as Hits@Majority (i.e., Exact Matching). An algorithmic description is given in Algo. 1. We report the results of UL2-20B and OPT-30B in Tab. 4. We can see that "No Recit" and "Hits@20-Path" account for the majority of the errors, meaning that the QA module performs quite well (if the correct answer appears in one of the recitation passages, it will be extracted by the QA module in most of the cases), and the main bottleneck still lies in the recitation quality and answer aggregation strategies.

We also perform a per-path error analysis, i.e., how many questions can be answered correctly (or not) when the recitation exactly contains (or not) the answer tokens. The results are shown in Tab. 5. We can see that around $7\% \sim 10\%$ questions have the correct recitation but cannot produce the correct answer, while around 12% questions do not have the correction recitation but can be answered correctly anyway.

## 5 CONCLUSION & DISCUSSION

In this paper, we propose a novel recitation-augmented generation framework to improve language models' performance in the closed-book question-answering setting. We hypothesize that for knowledge-intensive NLP tasks, encouraging the model to explicitly recite a specific knowledge source would be helpful in augmenting its memory. In addition, we found that diversifying the recitation process can be beneficial as well since usually there exists multiple knowledge sources that could be used to answer the same question. We show promising results over three large language models and across three different closed-book QA datasets, demonstrating the effectiveness of our proposed recite-and-answer approach.

One limitation of our method is that updating time-sensitive knowledge for a pure LLM-based method requires training or fine-tuning the LLMs on the new corpus, which can be costly. For future work, we plan to further validate the effectiveness of recitation-augmented generation for other knowledge-intensive NLP tasks in the closed-book setting, such as fact checking.

## ACKNOWLEDGEMENT

We thank the support and feedback of many people from Google Brain team and the constructive suggestions from the anonymous reviewers.

## ETHICS STATEMENT

The goal of this paper is to use recitation as an intermediate step to generate more accurate factual knowledge in the model's outputs. Therefore, our method should in principle improve the faithfulness of the LLM systems. However, unlike retrieval-augmented generation (RAG) models that can use external trustworthy corpus, all the intermediate steps in RECITE is generated by the LLM itself, RECITE may further enhance the existing biases in the LLMs' model weights compared to RAG.

## REPRODUCIBILITY STATEMENT

**Model weights** The model weights of two LLMs used in our experiments, i.e., UL2-20B (Tay et al., 2022a) and OPT-30B (Zhang et al., 2022), are publicly released through GCP bucket (`gs://scenic-bucket/ul2`) and Github (`https://github.com/facebookresearch/metaseq`), respectively. The Codex (`code-davinci-002`) model is publicly available through API calls (`https://beta.openai.com/examples/`).

**Evaluation datasets** The three evaluation datasets used in our experiments (Natural Questions[2], TriviaQA[3], and HotpotQA[4]) are all publicly accessible.

**Prompts** We provide all the used prompts in the appendix.

**Source code** Though the prompt examples in the appendix should be enough to reproduce all the results in our paper, we open-source all the evaluation code at `https://github.com/Edward-Sun/RECITE`.

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

# A    ILLUSTRATIONS OF PROMPTS AND LANGUAGE MODEL OUTPUTS

Fig. 7-15 illustrate the evidence-recitation, question-answering prompt that we used for Natural Questions, TriviaQA, and HotpotQA dataset. We also provide the example sampled recitations for these datasets. Notice that for Natural Questions, we use the "long answer" annotation as the recitation in the prompt, while for the other two datasets, we manually compose a few recitation passages based on web search.

# B    PRINCIPLES OF PROMPT DESIGNS

We mainly follow Chowdhery et al. (2022) and use two new line symbols "\n\n" as the separator between different components within exemplars, and use three new line symbol "\n\n\n" as the separator between different exemplars.

For the UL2 (Tay et al., 2022a) model, since its original SentencePiece (Kudo & Richardson, 2018) vocabulary does not encode the new line symbol "\n", we instead use " ; " to replace "\n" as the separator in all the prompts.

# C    DETAILS OF PASSAGE HINT-BASED FINE-TUNING

For Natural Questions (Kwiatkowski et al., 2019) dataset, we assume that we have top-retrieved Wikipedia pages for the unannotated queries issued to the Google search engine by multiple user. We collect the passages in these pages as a corpus, and use the rule to annotate the hints of these passages.

To make a fair comparison with prompting-based models in both 5-shot and 64-shot, we only use 5 paired "long answer"-question exemplars as the prompt to generate the synthetic question for the sampled passages from the Wikipedia hint-passage corpus, and thus construct the synthetic question-hint-passage paired fine-tuning data.

We train PaLM in the constructed corpus for 10,000 steps with a batch size of 64, which takes approximately 1 day in 64 TPUv4 chips[5]. The fine-tuned model can be used for passage hint-diversified recitation without any further prompts.

# D    PRE-TRAINED LANGUAGE MODELS

**PaLM**    PaLM is a family of densely activated decoder trained on the language modeling objective. It has strong capabilities in in-context few-shot learning, multilingual, as well as reasoning tasks. In this paper, we use the PaLM model with 62B parameters.

**UL2**    UL2 (Unifying Language Learning, Tay et al. 2022a) is an encoder-decoder model trained on a mixture of denoising tasks in a unified framework. In this paper, since we mainly focus on the in-context learning ability of language models, we use UL2-20B in the S-Denoiser mode (i.e., pre-trained with the prefix language modeling)[6].

**OPT**    OPT (Open Pre-trained Transformer language model, Zhang et al. 2022) is a family of recently released open-source densely activated language model that aims to re-reproduce comparable results as GPT-3 (Brown et al., 2020). We use the 30B one[7] in this paper.

**Codex**    Codex (Ouyang et al., 2022; Chen et al., 2021) is a variant of GPT-3 model (Brown et al., 2020) that can understand code. We use the public OpenAI API[8] to access the conditional generation outputs of the code-davinvi-002 model in this paper.

---

[5] https://cloud.google.com/tpu/docs/v4-users-guide

[6] This can be achieved by append the "[NLG]" and "[extra_id_0]" token to the beginning and the end of the prompt.

[7] https://github.com/facebookresearch/metaseq

[8] https://openai.com/api/

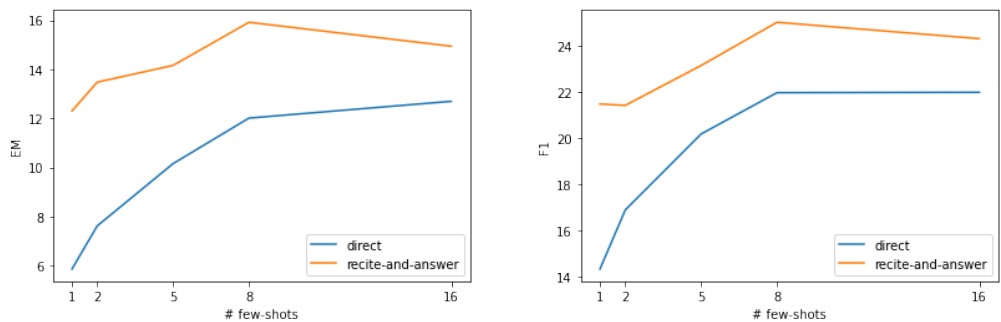

Figure 5: Nature Questions EM/F1 on UL2-20B with different # of shots.

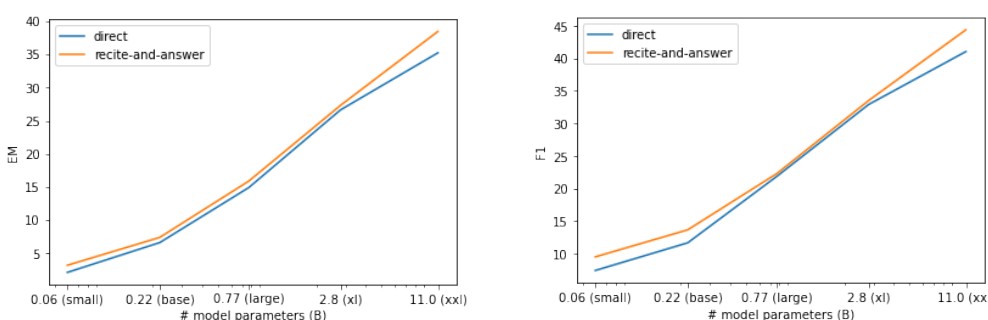

Figure 6: TriviaQA EM/F1 on 5-shot FLAN-T5 with different model sizes.

---

**Algorithm 1** Per-question Error Analysis

---

**Require:** $N$ ground-truth answer labels $\{A_i\}_{i=1}^{N}$ for a single question
**Require:** recitations $\{R_i\}_{i=1}^{K}$ and answer predictions $\{P_i\}_{i=1}^{K}$ from $K$ paths
    normalized_answers $\leftarrow \{\mathrm{normalize}(A_i)\}_{i=1}^{N}$
    **if** $\mathrm{normalize}(\mathrm{majority\_vote}(\{P_i\}_{i=1}^{K})) \in$ normalized_answers **then**
        **return** "Hits@Majority"
    **else if** $\mathrm{Any}(\{\mathrm{normalize}(P_i) \in \text{normalized\_answers}\}_{i=1}^{K})$ **then**
        **return** "Hits@20 − Path."
    **else if** $\mathrm{Any}(\{\mathrm{Any}(\{\text{normalized\_answers}_j \in \mathrm{normalize}(R_i)\}_{i=1}^{K})\}_{j=1}^{N})$ **then**
        **return** "Hits@20 − Recit."
    **else**
        **return** "NoRecit."
    **end if**

---

# E ANALYSIS OF THE NUMBER OF EXAMPLES IN FEW-SHOT LEARNING

We analyze the influence of the number of shots on Natural Questions dataset for standard-prompting model and our recite-and-answer model in Fig. 5. We can see that recite-and-answer prompting achieves consistent improvement over standard prompting, while the largest improvement is achieved in the 1-shot setting.

# F ANALYSIS OF MODEL SIZES ON INSTRUCTION-FINETUNED LANGUAGE MODELS

Instruction-finetuned language models (Sanh et al., 2021; Bach et al., 2022; Wei et al., 2021; Chung et al., 2022) trained on a collection of datasets phrased as instructions has been shown to improve

Table 6: Robustness evaluation of UL2 on TriviaQA with different few-shot exemplars over 5 random seeds.

|  | EM$_{(5)}$ | F1$_{(5)}$ |
|---|---|---|
| Standard (direct) | 48.42 ($\pm$ 0.71) | 53.85 ($\pm$ 0.57) |
| RECITE (5-path) | 49.75 ($\pm$ 0.50) | 54.78 ($\pm$ 0.46) |
| RECITE (20-path) | **52.68** ($\pm$ 0.62) | **58.05** ($\pm$ 0.58) |

model performance and generalization to unseen tasks. They tend to show better zero-shot or few-shot performance under the same model sizes compared to their vanilla LM pre-trained counterparts. We analyze the performance on one representative instruction-finetuned language model, FLAN-T5 (Chung et al., 2022), on TriviaQA dataset for standard-prompting model and our recite-and-answer model in Fig. 6. We can see that recite-and-answer prompting achieves consistent improvement over standard prompting, while the largest improvement is achieved in the largest "xxl (11B)" setting.

# G LANGUAGE MODEL HYPERPARAMETERS

We report the hyperparameters of the LLMs we used in Tab. 7.

Table 7: The model hyper-parameters of the large language models used in our experiments. Note that for Codex (i.e., `code-davinci-002`), the details (including size) are unknown, so we report the hyperparameters of GPT-3 as a common educational guess.

| Model | Type | # of Layers | # of Heads | $d_{\text{model}}$ | # of Parameters (B) |
|---|---|---|---|---|---|
| PaLM-62B | decoder-only | 64 | 32 | 8192 | 62.50 |
| UL2-20B | encoder-dencoder | 32 / 32 | 16 | 4096 | 19.46 |
| OPT-30B | decoder-only | 48 | 56 | 7168 | 29.97 |
| Codex* | decoder-only | 96 | 96 | 12288 | 175.0 |

> Question: where did united states drop the atomic bomb
>
> The answer to the above question can be found in the following Wikipedia page, section, and paragraph or table:
>
> Answer: During the final stage of World War II, the United States detonated two nuclear weapons over the Japanese cities of Hiroshima and Nagasaki on August 6 and 9, 1945, respectively. The United States dropped the bombs after obtaining the consent of the United Kingdom, as required by the Quebec Agreement. The two bombings killed at least 129,000 people, most of whom were civilians. They remain the only use of nuclear weapons in the history of warfare.
>
>
> Question: who sells the most oil in the world
>
> The answer to the above question can be found in the following Wikipedia page, section, and paragraph or table:
>
> Answer: The top three producers have in recent history been (alphabetically) Russia, Saudi Arabia, and the United States. Each of these countries experienced major production declines at different times in the past, but since 2014 all three have been producing near their peak rates of 9 to 11 million barrels per day. Saudi Arabia and Russia also top the list of oil exporting countries, while Russia is not a member of OPEC. The monthly U.S. oil production reached 10.07 million b/d in November 2017, the highest monthly level of crude oil production in U.S. history. In early 2018, the U.S. government forecast the country will become a net oil and gas exporter by 2022, the first time since 1953.

Figure 7: Two sampled evidence-recitation exemplars in Natural Questions (NQ) dataset. Notice that in NQ, we can directly use the "long answer" annotation as the recitation demonstrations.

Question: when did day light savings start in the us

Recitation: In the UK, the first Daylight Saving Time was first proposed by William Willett, an English builder and Liberal Party politician, in 1907, and after a public consultation his proposal was accepted in 1908. The first Daylight Saving in the UK was introduced on 25 March 1916, and the second on 27 March the following year. It was repealed in May 1919, but reintroduced in May 1919.

Recitation: Daylight saving time, also known as daylight savings, and several other variants, is the practice of advancing clocks during summer months by one hour so that evening daylight lasts one more hour. Usually, clocks are adjusted forward one hour near the start of spring and are adjusted backward in autumn. Historically, the time change was intended to conserve energy by making better use of the available daylight at the equinox.

Recitation: Daylight savings time in the United States is the practice of advancing clocks during summer months so that evening daylight lasts longer, while sacrificing normal sunrise times. The practice allows people to have more evening daylight and less morning daylight, which allows for more productivity and leisure.

Recitation: The United States began Daylight Saving Time in the year 1918, putting in effect a nationwide summer time arrangement, which had been put forward by the National Defense Act of 1916. Originally, Daylight Saving Time was started in the United States to save energy and reduce the cost of living. Currently, many countries have discontinued the practice, but some, such as the United States, observe it on a regional basis..

Figure 8: Four recitation passages sampled from UL2 for the same example question from Natural Questions dataset.

*Question*: Triggered by Rosa Parks' refusal to give up her seat, the public transportation system in what US city was devastated by a year long boycott of their busses?

*The answer to the above question can be found in the following Wikipedia page, section, and paragraph:*

*Answer*: On December 1, 1955, in Montgomery, Alabama, Parks rejected bus driver James F. Blake's order to vacate a row of four seats in the "colored" section in favor of a White passenger, once the "White" section was filled.

*Question*: What was Beethoven's last symphony?

*The answer to the above question can be found in the following Wikipedia page, section, and paragraph:*

*Answer*: The year 1823 saw the completion of three notable works, all of which had occupied Beethoven for some years, namely the Missa solemnis, the Ninth Symphony and the Diabelli Variations.

*Question*: In which 1972 John Boorman film is a leading character, played by Ned Beatty, raped by a 'Hillbilly'?

*The answer to the above question can be found in the following Wikipedia page, section, and paragraph:*

*Answer*: In 1972, Beatty made his film debut as Bobby Trippe in Deliverance, starring Jon Voight and Burt Reynolds, and set in northern Georgia. Beatty's character is forced to strip at gunpoint by two mountain men who humiliate and rape him, a scene so shocking that it is still referred to as a screen milestone.

*Question*: Which bridge crossing The River Thames did Queen Elizabeth II open on 17th March 1973?

*The answer to the above question can be found in the following Wikipedia page, section, and paragraph:*

*Answer*: Queen Elizabeth II of the United Kingdom opens the new London Bridge.

*Question*: "The song ""My Kind Of Town"", written by Sammy Cahn and Jimmy Van Heusen in 1964, was about which city?"

*The answer to the above question can be found in the following Wikipedia page, section, and paragraph:*

*Answer*: "My Kind of Town" made a minor appearance on the U.S. pop charts, reaching #110 in 1964. It was the second of two charting songs about Chicago recorded by Sinatra. The other was "Chicago (That Toddlin' Town)" from 1957, which reached U.S. #84.

Figure 9: The 5-shot prompt we used for performing evidence-recitation on TriviaQA dataset.

Question: Which physicist's principle asserts that the momentum & position of a particle cannot both be precisely determined at the same time?

Recitation: In 1935, Heisenberg (along with his assistants Walther Gerlach and Walther von Braun) published a paper in which they described a "principle of uncertainty" that would later be dubbed the Heisenberg uncertainty principle. The principle asserts that the momentum and position of a particle cannot both be precisely determined.

Recitation: One of the consequences of the principle of special relativity is that the position and momentum of a moving object cannot both be measured simultaneously with high precision.

Recitation: The uncertainty principle states that it is impossible to know both the exact momentum and the exact position of an object (particle) at the same time, that is, the momentum and position are incompatible and, therefore, complementary. The uncertainty principle is one of the fundamental results of quantum mechanics, and is often regarded as one of its most surprising consequences.

Recitation: It states that the following two statements cannot both be true: the position of a particle cannot be both precisely determined and known exactly, and the momentum of a particle cannot be both precisely measured and known exactly.

Figure 10: Four recitation passages sampled from UL2 for the same example question from Trivia-aQA dataset.

*Q:* Triggered by Rosa Parks' refusal to give up her seat, the public transportation system in what US city was devastated by a year long boycott of their busses?

*A:* Montgomery, Al

*Q:* What was Beethoven's last symphony?

*A:* 9th

*Q:* In which 1972 John Boorman film is a leading character, played by Ned Beatty, raped by a 'Hillbilly'?

*A:* Deliverance

*Q:* Which bridge crossing The River Thames did Queen Elizabeth II open on 17th March 1973?

*A:* London Bridge

*Q:* "The song ""My Kind Of Town"", written by Sammy Cahn and Jimmy Van Heusen in 1964, was about which city?"

*A:* Chicago

Figure 11: The 5-shot prompt we used for performing question-answering on TriviaQA dataset.

---

*Question:* Which magazine was started first Arthur's Magazine or First for Women?

*The answer to the above question can be found in the following two Wikipedia page, section, and paragraphs:*

*Answer 1:* Arthur magazine was a bi-monthly periodical that was founded in October 2002, by publisher Laris Kreslins and editor Jay Babcock.

*Answer 2:* First for Women is a woman's magazine published by Bauer Media Group in the USA. The magazine was started in 1989.

*Question:* The Oberoi family is part of a hotel company that has a head office in what city?

*The answer to the above question can be found in the following two Wikipedia page, section, and paragraphs:*

*Answer 1:* P.R.S. Oberoi is the current chairman of The Oberoi Group.

*Answer 2:* The Oberoi Group is an award-winning luxury hotel group with its head office in New Delhi, India.

*Question:* What nationality was James Henry Miller's wife?

*The answer to the above question can be found in the following two Wikipedia page, section, and paragraphs:*

*Answer 1:* In 1967, Miller married his fifth wife, Japanese born singer Hoki Tokuda.

*Answer 2:* Hoki Tokuda is an Japanese actress, known for Blind Woman's Curse (1970), The Abalone Girls (1965) and Nippon Paradise (1964).

*Question:* The Dutch-Belgian television series that "House of Anubis" was based on first aired in what year?

*The answer to the above question can be found in the following two Wikipedia page, section, and paragraphs:*

*Answer 1:* House of Anubis is a mystery television series developed for Nickelodeon based on the Dutch–Belgian television series Het Huis Anubis.

*Answer 2:* Het Huis Anubis (English: The House of Anubis) is a Dutch-Belgian children's television mystery drama. It first aired in September 2006.

---

Figure 12: The 4-shot prompt we used for performing multiple-evidence recitation on HotpotQA dataset.

---

*Q:* Which magazine was started first Arthur's Magazine or First for Women?

*A: The answer is* Arthur's Magazine.

*Q:* The Oberoi family is part of a hotel company that has a head office in what city?

*A: The answer is* Delhi.

*Q:* What nationality was James Henry Miller's wife?

*A: The answer is* American.

*Q:* The Dutch-Belgian television series that "House of Anubis" was based on first aired in what year?

*A: The answer is* 2006.

---

Figure 13: The 4-shot prompt we used for performing question-answering on HotpotQA dataset.

Question: Are Local H and For Against both from the United States?

Answer 1: Local H is an American band from San Diego, California.

Answer 2: For Against is an American punk band from Los Angeles, California.

Answer 1: Local H is a rock band from Los Angeles, California.

Answer 2: For Against is an American rock band from Phoenix, Arizona.

Answer 1: Local H is an American rock band from Austin, Texas.

Answer 2: For Against is a rock band from Nashville, Tennessee.

Answer 1: Local H is an American rock band from Chicago, Illinois.

Answer 2: For Against is an indie rock band from New York City, New York.

Figure 14: Four recitation sampled from UL2 for the same example question from HotpotQA dataset.

*Q:* Which magazine was started first Arthur's Magazine or First for Women?

*A:* Arthur's Magazine started in 1844. First for Women started in 1989. So Arthur's Magazine was started first. The answer is Arthur's Magazine.

*Q:* The Oberoi family is part of a hotel company that has a head office in what city?

*A:* The Oberoi family is part of the hotel company called The Oberoi Group. The Oberoi Group has its head office in Delhi. The answer is Delhi.

*Q:* What nationality was James Henry Miller's wife?

*A:* James Henry Miller's wife is June Miller. June Miller is an American. The answer is American.

*Q:* The Dutch-Belgian television series that "House of Anubis" was based on first aired in what year?

*A:* "House of Anubis" is based on the Dutch–Belgian television series Het Huis Anubis. Het Huis Anubis is firstaired in September 2006. The answer is 2006.

Figure 15: The 4-shot prompt we used for the Chain-of-Thought (Wei et al., 2022) baseline on HotpotQA dataset. The prompt is taken from (Wang et al., 2022a).

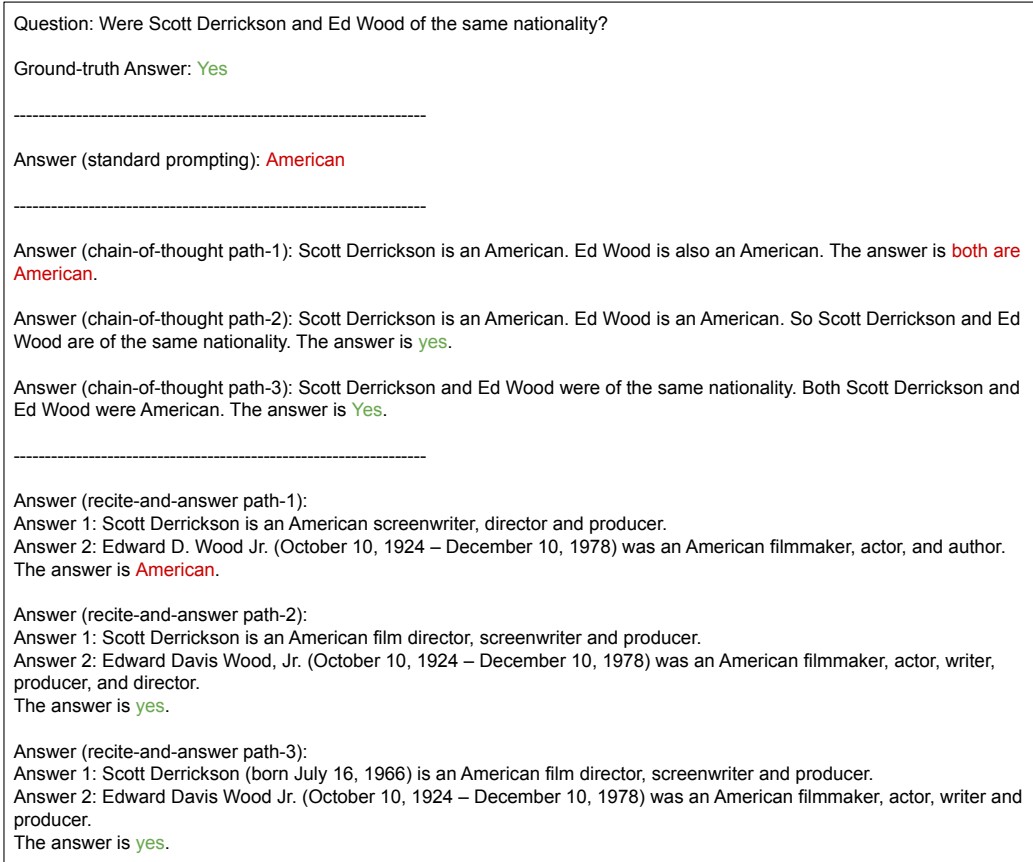

Figure 16: Qualitative comparisons between standard prompting, chain-of-thought, and recite-and-answer on HotpotQA evaluation example (I).

Question: The director of the romantic comedy "Big Stone Gap" is based in what New York city?

Ground-truth Answer: Greenwich Village, New York City

---------------------------------------------------------------

Answer (standard prompting): New York City

---------------------------------------------------------------

Answer (chain-of-thought path-1): Adriana Trigiani is the director of the romantic comedy Big Stone Gap. Adriana Trigiani is based in New York City. The answer is New York City.

Answer (chain-of-thought path-2): The director of the romantic comedy "Big Stone Gap" is Adriana Trigiani. Adriana Trigiani is based in New York City. The answer is New York City.

Answer (chain-of-thought path-3): The director of "Big Stone Gap" is Adriana Trigiani. Adriana Trigiani lives in Manhattan. The answer is Manhattan.

---------------------------------------------------------------

Answer (recite-and-answer path-1):
Answer 1: Big Stone Gap is a 2014 American romantic comedy film directed by Adriana Trigiani.
Answer 2: Adriana Trigiani is an American author, playwright, filmmaker, and entrepreneur. She lives in Greenwich Village, New York City.
The answer is Greenwich Village.

Answer (recite-and-answer path-2):
Answer 1: Big Stone Gap is an American romantic comedy film directed by Adriana Trigiani and written by Trigiani and Andrea Berloff.
Answer 2: Adriana Trigiani is an American bestselling author, filmmaker, and playwright. She lives in Greenwich Village, New York.
The answer is Greenwich Village.

Answer (recite-and-answer path-3):
Answer 1: Adriana Trigiani is an American writer and filmmaker.
Answer 2: Big Stone Gap is a 2014 American romantic comedy film written and directed by Adriana Trigiani, based on her 2000 debut novel of the same name.
The answer is New York City.

Figure 17: Qualitative comparisons between standard prompting, chain-of-thought, and recite-and-answer on HotpotQA evaluation example (II).

