# OpenReview forum: "Recitation-Augmented Language Models"
_ICLR.cc/2023/Conference — ICLR 2023 poster_

### Official Review · Reviewer_m3Mw · 2022-10-15

**Confidence:** 5
**Correctness:** 3
**Technical Novelty And Significance:** 3
**Empirical Novelty And Significance:** Not applicable
**Recommendation:** 5

**Clarity, Quality, Novelty And Reproducibility:**

Clarity: Very clear
Quality: Very high quality
Novelty: somehow novel
Reproducibility: 2 out of three LM are available for the research community.


**Strength And Weaknesses:**

Strength
- Retrieval free QA model are an important topic to study, and RECITE enable midsize LMs to achieve competitive performance in the close-book QA.

Weaknesses
- As also mentioned is Limitation, the main drawback is the the updated of the model with new knowledge. This is even more evident when compared to simple DB retrieval in Table 5, where BM25 performs better that LM-Recitation 5-shot with 20-path (the best proposed model).

Questions/Suggestion:
- What is the size of the index + docs (compressed) of BM25? is the size larger or smaller of the model parameter? if the size of the DB is larger of the model parameter, why not constrain it (e.g., randomly removing articles) and check if the performance in Table 5 changes?
- What is the inference speed of LM-Recitation K-shot with t-path? How's compared to BM25?
- Does scale matter? what's the performance of larger LM (PaLM-500B or GPT-3/Instruct-GPT3) or smaller (GPT-J 6B)?
- Figure 4 shows the performance difference by sampling more self-consistency paths, but what append if the model samples more responses for the direct method (or chain-of-thoughts)?

**Summary Of The Paper:**

The authors presents RECITE, a novel prompting Language Model (LM) paradigm to improve know knowledge-intensive NLP tasks. The main idea is to first let the language model to generate knowledge via prompting (knowledge-recitation) and then to let the language model solve the task (task-execution). The author evaluate the performance of the proposed approach with three close book QA tasks (Natural Questions, TriviaQA, and HotpotQA) and three LMs (In-house LM (62B), UL2 (20B), and OPT (30B)). The results shows better performance than direct prompting and chain-of-thoughts, and comparable performance to retrieval methods (BM25).

**Summary Of The Review:**

The paper propose a two step approach to improve knowledge-intensive NLP tasks by using only LMs (no external retriever).  However the model performance are a bit lagging compare to simple BM25 + LMs, plus the clear weakness of updated knowledge.

---

> ### Author Response · Authors · 2022-11-18
> **Response to Reviewer m3Mw**
>
> We thank the reviewer for their time, insightful comments, and questions. We have provided our responses below.
>
> > As also mentioned is Limitation, the main drawback is the the updated of the model with new knowledge. This is even more evident when compared to simple DB retrieval in Table 5, where BM25 performs better that LM-Recitation 5-shot with 20-path (the best proposed model).
>
> Yes we acknowledge that this is a major limitation. We do show in Section 3.2 that we can further incorporate an external corpus via fine-tuning the LLM, so updated knowledge can also be incorporated in this way.
>
> > What is the size of the index + docs (compressed) of BM25? is the size larger or smaller of the model parameter? if the size of the DB is larger of the model parameter, why not constrain it (e.g., randomly removing articles) and check if the performance in Table 5 changes?
>
> As reported in the paper, we used the ``enwiki-paragraphs’’ corpus in the pyserini package. The compressed size is 16.5 GB, which is smaller than the model parameters. Though, we don’t think it’s fair to directly compare CBQA and OBQA performance.
>
> > What is the inference speed of LM-Recitation K-shot with t-path? How's compared to BM25?
>
> It’s slower than BM25. Though, we don’t think it’s fair to directly compare CBQA and OBQA performance. Note that BM25 requires an external corpus, i.e., we know which corpus we should retrieve from. This might not be true given a random question while our RECITE model can still handle this.
>
> > ​​Does scale matter? what's the performance of larger LM (PaLM-500B or GPT-3/Instruct-GPT3) or smaller (GPT-J 6B)?
>
> We presented new results on Codex-(with 175B params) in Table 1, where recite-and-answer shows significant improvement on NQ and HotpotQA:
>
> | EM / F1  | Codex (direct) | Codex (chain-of-thought) | Codex (recite-and-answer) |
> |----------|----------------|--------------------------|---------------------------|
> | NQ       | 30.96 / 42.77  | -                        | **33.98 / 47.48**         |
> | TriviaQA | 83.40 / 87.12  | -                        | **83.50 / 87.97**         |
> | HotpotQA | 29.10 / 39.86  | 33.87 / 45.27            | **37.60 / 48.62**         |
>
> We also provide additional analysis of increasing model sizes (small, base, large, xl, xxl) on an instruction-finetuned language model (FLAN-T5) in Appendix F and Figure 6. We show that recite-and-answer prompting achieves consistent improvement over standard prompting, while the largest improvement is achieved in the largest “xxl (11B)” setting.
>
> > Figure 4 shows the performance difference by sampling more self-consistency paths, but what append if the model samples more responses for the direct method (or chain-of-thoughts)?
>
> In our paper, both chain-of-thought and recite-and-answer use self-consistency with 20 paths. The self-consistency technique does not apply to direct prompting (in direct prompting, sample-and-vote is the same as greedy decoding). It works amazingly with the chain of thought prompting because it marginalizes all the latent reasoning paths to obtain a full probability of the final answer.

---

> > ### Comment · Reviewer_m3Mw · 2022-12-01
> > **Re:**
> >
> > Thanks for the further experiments and explanation.
> >
> > As I mentioned in my review, the idea is definitely interesting but lack of a proper motivation. Practicality, the idea is quite unreasonable compared to a simple baseline retrieval model +LM. For instance:
> > - finetuning to add knowledge is possible, but much more expensive than simply adding the doc to the DB
> > - the size of the index + docs is smaller than the model, which is actually very important any scenarios.
> > - Generating 20 paths, is very slow especially for LLM, and also expensive since the model need to use large GPUs to fit the batch or more API credits.
> >
> > > It’s slower than BM25. Though, we don’t think it’s fair to directly compare CBQA and OBQA performance. Note that BM25 requires an external corpus, i.e., we know which corpus we should retrieve from. This might not be true given a random question while our RECITE model can still handle this.
> >
> > LLM requires a corpus too to train or to update the knowledge. And what are actually random question? I guess you refer to OOD question, then it depends, if the question is not covered in the LLM corpus, it cannot be answer anyway. On the other hand, if the retriever returns empty results, or low confidence hypothesis, it could be used as an OOD detector. Anyhow, this should be somehow quantified before claiming that RECITE can handle these scenarios.

---

> > > ### Author Response · Authors · 2022-12-01
> > > **Further clarification**
> > >
> > > We thank the reviewer for the further feedback.
> > >
> > > > Practicality, the idea is quite unreasonable compared to a simple baseline retrieval model +LM.
> > >
> > > This paper proposes a new paradigm to help LLMs generate more accurate factual knowledge
> > > *without* retrieving from an external corpus. In the literature, it is called the closed-book setting, which should not be directly compared to the open-book results.
> > >
> > > > LLM requires a corpus too to train or to update the knowledge. And what are actually random question?
> > >
> > > Recall that the open-book setting is in analogy as taking an open-book exam where students can search over an external knowledge corpus, while the closed-book setting is in analogy as taking a closed-book exam. Therefore, while retrieval in the open-book setting can help when the answer can be found in a given corpus (e.g., how these QA datasets are constructed), the recitation and closed-book QA focuses on utilizing the knowledge in the LLM’s weights, and can answer random questions (e.g., with a reasonable/educated guess, imagining how human taking closed-book exams).

---

> > > > ### Comment · Reviewer_m3Mw · 2022-12-01
> > > > **Re: Further Clarification**
> > > >
> > > > > This paper proposes a new paradigm to help LLMs generate more accurate factual knowledge without retrieving from an external corpus. In the literature, it is called the closed-book setting, which should not be directly compared to the open-book results.
> > > >
> > > > Sure, I am familiar with that, and I want to further acknowledge the novelty and interestedness of the proposed recitation approach. However, given the lack in performance (vs simple baselines) and the necessary computational cost (bigger in size and some X slower than the baselines), is hard for justify the proposed approach.
> > > >
> > > > > Recall that the open-book setting is in analogy as taking an open-book exam where students can search over an external knowledge corpus, while the closed-book setting is in analogy as taking a closed-book exam. Therefore, while retrieval in the open-book setting can help when the answer can be found in a given corpus (e.g., how these QA datasets are constructed), the recitation and closed-book QA focuses on utilizing the knowledge in the LLM’s weights, and can answer random questions (e.g., with a reasonable/educated guess, imagining how human taking closed-book exams).
> > > >
> > > > Thanks for the clarification.

---

> > > > > ### Author Response · Authors · 2022-12-06
> > > > > **Re: Re: Further Clarification**
> > > > >
> > > > > > “However, given the lack in performance (vs simple baselines) and the necessary computational cost (bigger in size and some X slower than the baselines), is hard for justify the proposed approach.”
> > > > >
> > > > > We thank the reviewer for the clarification. We understand the reviewer thinks the retrieval-augmented generation is an important direction for further improving LLM’s performance on knowledge-intensive tasks. However, note that in some emergent powerful dialogue APIs, such as ChatGPT (https://chat.openai.com), it is explicitly enforced that they are “not able to browse the internet or access information outside of my training data, which only goes up until 2021.” and “not able to provide current information or answer questions about events or people outside of that time period”. In this case, the recitation-augmented generation can be an effective way to directly improve the performance through simple prompting.
> > > > >
> > > > > P.S.
> > > > >
> > > > > Though it is not possible to evaluate a large quantity of QA examples in the web API, the reviewer may try the following direct answering prompt and recite-and-answer prompt for some random questions, where we found the results are quite impressive (illustrated with the first HotpotQA example in the appendix as below).
> > > > >
> > > > > Direct:
> > > > > ---
> > > > >
> > > > > Input:
> > > > > ```
> > > > > Make an educated guess to the following question:
> > > > >
> > > > > Were Scott Derrickson and Ed Wood of the same nationality?
> > > > > ```
> > > > > Output:
> > > > > ```
> > > > > It is likely that Scott Derrickson and Ed Wood were not of the same nationality.
> > > > > ```
> > > > >
> > > > > Recite-and-answer:
> > > > > ---
> > > > >
> > > > > Input:
> > > > > ```
> > > > > Generate two relevant Wikipedia paragraphs, and make an educated guess to the following question:
> > > > >
> > > > > Were Scott Derrickson and Ed Wood of the same nationality?
> > > > > ```
> > > > > Output:
> > > > > ```
> > > > > Scott Derrickson is an American film director, screenwriter, and producer. He is best known for his work in the horror genre, with films such as The Exorcism of Emily Rose, Sinister, and Doctor Strange. Derrickson was born in Denver, Colorado, and grew up in the suburban town of Littleton. He attended the University of Southern California's School of Cinematic Arts and began his career as a screenwriter in the mid-1990s.
> > > > > Ed Wood was an American filmmaker, actor, and author. He is often considered one of the worst directors in film history due to his low-budget productions and unconventional storytelling techniques. Wood was born in Poughkeepsie, New York, and grew up in Los Angeles. He began his career as a screenwriter in the 1940s, but quickly moved into directing, producing, and acting in his own films.
> > > > > Based on the information provided, it is likely that both Scott Derrickson and Ed Wood were of the same nationality, as they were both born and raised in the United States.
> > > > > ```

---

> > > > > > ### Comment · Reviewer_m3Mw · 2022-12-06
> > > > > > **Re: Re: clarification.**
> > > > > >
> > > > > > FYI:
> > > > > >
> > > > > > Direct question to ChatGPT:
> > > > > > ```
> > > > > > Were Scott Derrickson and Ed Wood of the same nationality?
> > > > > > ```
> > > > > > ```
> > > > > > Scott Derrickson and Ed Wood were both American filmmakers. Derrickson was born in Denver, Colorado, and Wood was born in Poughkeepsie, New York. They both made a number of notable films, but their styles and approaches to filmmaking were quite different. Derrickson is known for his work in the horror genre, while Wood is best known for his low-budget, cult films.
> > > > > > ```
> > > > > >
> > > > > > Plus, GPT-3 davinci-003 CoT:
> > > > > >
> > > > > > ```
> > > > > > Were Scott Derrickson and Ed Wood of the same nationality?
> > > > > >
> > > > > > let's think step by step.
> > > > > >
> > > > > > Scott Derrickson is an American film director, producer, and screenwriter.
> > > > > >
> > > > > > Ed Wood was an American director, writer, producer, and actor.
> > > > > >
> > > > > > Therefore, Scott Derrickson and Ed Wood were both of the same nationality, which is American.
> > > > > > ```

---

> > > > > > > ### Author Response · Authors · 2022-12-06
> > > > > > > **Re: Re: Re: clarification**
> > > > > > >
> > > > > > > We thank the reviewer for showing a direct question result. We have tried this prompt (null prompt) before. However, we found that for many knowledge-intensive questions, null prompt will result in
> > > > > > >
> > > > > > > > I'm sorry, but I'm not able to browse the internet or access current information. I am a large language model trained by OpenAI, and my knowledge is based on the text that I have been trained on. My training data only goes up until 2021, so I would not be able to provide information about events or people after that time. Is there something else I can help you with?
> > > > > > >
> > > > > > > While our prompt above, i.e., "Make an educated guess to the following question:", can help the ChatGPT model generate an answer rather than refuse to answer it.
> > > > > > >
> > > > > > > Other example questions can be found in the following links:
> > > > > > >
> > > > > > > http://curtis.ml.cmu.edu/datasets/hotpot/hotpot_dev_fullwiki_v1.json
> > > > > > > http://curtis.ml.cmu.edu/datasets/hotpot/hotpot_test_fullwiki_v1.json

---

> > > > > > > > ### Comment · Reviewer_m3Mw · 2022-12-06
> > > > > > > > **Re: Re: Re: clarification**
> > > > > > > >
> > > > > > > > Ok, I feel this thread is not relevant anymore to the review. Out of curiosity, try the refresh button a couple of time in ChatGPT.

---

> > > > > > > ### Author Response · Authors · 2022-12-06
> > > > > > > **Re: Re: clarification**
> > > > > > >
> > > > > > > We thank the reviewer for further showing the GPT-3 results. For a comprehensive comparison between recite-and-answer and chain-of-thought on GPT-3, please refer to Table 1 for the quantitative comparison.

---

> > > > > > > > ### Comment · Reviewer_m3Mw · 2022-12-06
> > > > > > > > **Re: Re: clarification**
> > > > > > > >
> > > > > > > > Thanks for the pointer, I guess you mean table 16/17.
> > > > > > > >
> > > > > > > > Anyhow, cherry peaking examples to make your point it's not ideal, especially with newer models.
> > > > > > > >
> > > > > > > > That's said, thanks again for pointing out these examples and valid points.

---

> > > > > > > > > ### Author Response · Authors · 2022-12-06
> > > > > > > > > **Re: Re: Re: clarification**
> > > > > > > > >
> > > > > > > > > In Table 1, we quantitatively (numerically) compared recite-and-answer and chain-of-thought with Codex (GPT-3, code-davinci-002) on the first 1,024 evaluation examples of HotpotQA.

---

> > > > > > > > > > ### Comment · Reviewer_m3Mw · 2022-12-06
> > > > > > > > > > **Re: Re: Re: clarification**
> > > > > > > > > >
> > > > > > > > > > Some points to make:
> > > > > > > > > > - in NQ and TriviaQA the improvement are marginal, and no results with CoT with 20 paths (which again is quite absurd).
> > > > > > > > > > - in HotpotQA are better but 1) code-davinci-002 is very different compared to text-davinci-002/3 and 2) very far from any existing SOTA with a retriever. Where 2) is the main point of my all review.

---

> > > > > > > > > > > ### Author Response · Authors · 2022-12-06
> > > > > > > > > > > **Re: Re: Re: Re: clarification**
> > > > > > > > > > >
> > > > > > > > > > > We thank the reviewer for the feedback. However, we would like to clarify that
> > > > > > > > > > >
> > > > > > > > > > > > no results with CoT with 20 paths (which again is quite absurd).
> > > > > > > > > > >
> > > > > > > > > > > To the best of our knowledge, there is no existing CoT work/prompts on the single-hop knowledge-intensive tasks such as NQ or TriviaQA.
> > > > > > > > > > >
> > > > > > > > > > > > in HotpotQA are better but ... 2) very far from any existing SOTA with a retriever.
> > > > > > > > > > >
> > > > > > > > > > > Please note that in this paper, we focus on knowledge-intensive NLP tasks **in the few-shot setting**. Therefore, it is not fair to compare RECITE with the models fine-tuned on the full training set.
> > > > > > > > > > >
> > > > > > > > > > > In addition, although not using exactly the same evaluation protocol, we would like to point out that Codex with recite-and-answer can achieve competitive performance with the current SOTA retrieval-augmented language model with the few-shot learning ability, i.e., Atlas [1]. We hope the comparison between Table 1 in our paper and Table 10 & Table 19 in [1] can partially address the reviewer's concerns about SOTA results (**in the few-shot setting**).
> > > > > > > > > > >
> > > > > > > > > > > [1]: Izacard, Gautier, Patrick Lewis, Maria Lomeli, Lucas Hosseini, Fabio Petroni, Timo Schick, Jane Dwivedi-Yu, Armand Joulin, Sebastian Riedel, and Edouard Grave. "Few-shot learning with retrieval augmented language models." arXiv preprint arXiv:2208.03299 (2022).

---

> > > > > > > > > > > > ### Comment · Reviewer_m3Mw · 2022-12-06
> > > > > > > > > > > > **Re: .... Re: clarification**
> > > > > > > > > > > >
> > > > > > > > > > > > > To the best of our knowledge, there is no existing CoT work/prompts on the single-hop knowledge-intensive tasks such as NQ or TriviaQA.
> > > > > > > > > > > >
> > > > > > > > > > > > Thanks for pointing this out. I was not aware of this details, for knowledge-intensive tasks, so the results in Table 1 for NQ, and Trivia, are also 20 paths? or do they use CoT?
> > > > > > > > > > > >
> > > > > > > > > > > > > In addition, although not using exactly the same evaluation protocol, we would like to point out that Codex with recite-and-answer can achieve competitive performance with the current SOTA retrieval-augmented language model with the few-shot learning ability, i.e., Atlas [1]. We hope the comparison between Table 1 in our paper and Table 10 & Table 19 in [1] can partially address the reviewer's concerns about SOTA results (in the few-shot setting).
> > > > > > > > > > > >
> > > > > > > > > > > > To be fair, Atlas is 11B model which performs on par or better in 60B model with recitation that you report in Table 1 (which btw I don't get it why more results are not reported in the paper).

---

> > > > > > > > > > > > > ### Author Response · Authors · 2022-12-06
> > > > > > > > > > > > > **Re: .... Re: clarification**
> > > > > > > > > > > > >
> > > > > > > > > > > > > We thank the reviewer for the discussion.
> > > > > > > > > > > > >
> > > > > > > > > > > > > > so the results in Table 1 for NQ, and Trivia, are also 20 paths? or do they use CoT?
> > > > > > > > > > > > >
> > > > > > > > > > > > > The standard prompting (direct) method in this paper only uses 1 path with greedy decoding, since "in direct prompting, sample-and-vote is the same as greedy decoding".
> > > > > > > > > > > > >
> > > > > > > > > > > > > > which btw I don't get it why more results are not reported in the paper
> > > > > > > > > > > > >
> > > > > > > > > > > > > This is because the evaluation protocols are different in the two papers (e.g., 4,5-shot v.s. 64-shot, first 1,024 examples v.s. full KILT...), and the Atlas model is not public.

---

> > > > > > > > > > > > > > ### Comment · Reviewer_m3Mw · 2022-12-06
> > > > > > > > > > > > > > **Re: .... Re: clarification**
> > > > > > > > > > > > > >
> > > > > > > > > > > > > > > The standard prompting (direct) method in this paper only uses 1 path with greedy decoding, since "in direct prompting, sample-and-vote is the same as greedy decoding".
> > > > > > > > > > > > > >
> > > > > > > > > > > > > > why not comparing with CoT 20 paths?
> > > > > > > > > > > > > >
> > > > > > > > > > > > > > > This is because the evaluation protocols are different in the two papers (e.g., 4,5-shot v.s. 64-shot, first 1,024 examples v.s. full KILT...), and the Atlas model is not public.
> > > > > > > > > > > > > >
> > > > > > > > > > > > > > I see, this can be a bit tricky indeed. Anyhow it can help the reader to contextualize the results.

---

> > > > > > > > > > > > > > > ### Author Response · Authors · 2022-12-06
> > > > > > > > > > > > > > > **Re: .... Re: clarification**
> > > > > > > > > > > > > > >
> > > > > > > > > > > > > > > We thank the reviewer for the discussion.
> > > > > > > > > > > > > > >
> > > > > > > > > > > > > > > > why not comparing with CoT 20 paths?
> > > > > > > > > > > > > > >
> > > > > > > > > > > > > > > For the multi-hop QA task, i.e., HotpotQA, the chain-of-thought prompt is taken from [1]. However, "to the best of our knowledge, there is no existing CoT work/prompts on the single-hop knowledge-intensive tasks such as NQ or TriviaQA". Therefore, for these tasks, we only compare recite-and-answer with standard prompting (direct) in this paper.
> > > > > > > > > > > > > > >
> > > > > > > > > > > > > > > [1]: Xuezhi Wang, Jason Wei, Dale Schuurmans, Quoc Le, Ed Chi, and Denny Zhou. Rationaleaugmented ensembles in language models. arXiv preprint arXiv:2207.00747, 2022a

---

### Official Review · Reviewer_HxiC · 2022-10-21

**Confidence:** 4
**Correctness:** 3
**Technical Novelty And Significance:** 3
**Empirical Novelty And Significance:** 3
**Recommendation:** 6

**Clarity, Quality, Novelty And Reproducibility:**

*Clarity* - Very clear

*Quality* - Very thorough experiments overall except for the experiments on passage hints. I would have liked other models being tested (weakness #1), but the experiments on the current set of tasks look good to me.

*Novelty* - Good novelty. The idea has similarity to chain-of-thought prompting, self-consistency prompting, and self-talk, but overall I think the idea is pretty new (especially in the context of large LMs and QA).

*Reproducibility* - Should be fully reproducible except the experiments on the in-house LM.

**Strength And Weaknesses:**

**Strengths**

1. This is an interesting idea, and an exciting way to incorporate the ideas of "chain-of-thought" prompting and "self-consistency" for open-domain QA tasks where the answer is a factoid.

2. The authors observe 2-6% improvements over standard direct prompting across all 3 tasks / 3 models. This is quite good and surprising for me --- I had thought the model would be able to answer the factoid question directly if it's able to generate a much longer paragraph containing the answer. It is interesting that the method requires multiple recitation paths to work (in Figure 4 performance is lower than standard prompting with just one recitation path). However, I think of this as a strength of the proposed method, since you cannot really use multiple paths if you are generating a direct answer (since the answer is so short, sampling doesn't make sense).

3. The authors perform several insightful analysis experiments discussing robustness to prompts, comparison to BM25 retrieval, and an error analysis.

**Weaknessess**

1. The paper would be much stronger with experiments on GPT3, Instruct-GPT3 (davinci-002), and larger language models (larger in-house LMs?). It's not really clear from the paper whether recitation helps with larger scale, which I think is important for the generalizability of the method [1]. This could work both ways --- I'm suspecting larger LMs will be better at both recitation and directly performing QA. I think experiments on InstructGPT [4], T0 [3] or FLAN [2, 7] will be especially interesting, since it's been fine-tuned on instructions / examples / human preferences.

2. A major advantage of retrieval augmented systems is their applicability on (1) tail distribution information; (2) generalization to information which was not present in the model's training set (like COVID for BERT). I think these are important limitations of the proposed method, and (1) is not really discussed (2 is just mentioned in the conclusion). Are most of the correct recitations cases which were seen often during training?

3. Overall, the performance of closed-book models in this paper seems to significantly lag behind recent few-shot retrieval-augmented systems [5, 6]. For instance, ATLAS [5] gets 42% on NQ with 64 examples and a smaller model, while the best number in this paper is 32% (5-10x larger model). While I agree that setting up retrieval is technically cumbersome, there are very good retrieval APIs available, which were utilized in [6] without any extra LM fine-tuning. Note that I do think it's incredible that closed book LMs are doing so well, but practically (from a performance stand-point) it may be better to just retrieve some text from the web rather than ask an LM to generate it with few-shot examples. Also, retrieval augmented LMs often have lesser parameters [5], so it's unclear which is a better method from an efficiency perspective.

4. I have mixed thoughts about the passage hints fine-tuning experiments, since it requires fine-tuning a large LM on Wikipedia data. Perhaps the performance gains are because of the dedicated fine-tuning on Wikipedia data for the recitation LM model (which makes it overfit to Wikipedia)? Did you remove the passages from the test set questions while doing this fine-tuning? Also I don't think enough experiments are done in the paper to justify its added complexity over vanilla LM-Recitation. I would suggest moving it to the appendix, or performing experiments on all 3 datasets / models to show its benefit.

[1] - https://twitter.com/_jasonwei/status/1526589104758042624
[2] - https://arxiv.org/abs/2109.01652
[3] - https://arxiv.org/abs/2110.08207
[4] - https://arxiv.org/abs/2203.02155
[5] - https://arxiv.org/abs/2208.03299
[6] - https://arxiv.org/abs/2203.05115
[7] - https://arxiv.org/abs/2210.11416

**Minor**

This paper is relevant to https://arxiv.org/abs/2004.05483 and https://arxiv.org/pdf/2110.08387.pdf, it would be great to cite them.

In Table 4 (LM-Recitation_5), why is the number for different from Table 1 (Recite and answer)? (16.34 in Table 1 vs 14.16 in Table 4)

**Summary Of The Paper:**

This paper presents a new prompt engineering algorithm for few-shot open-domain question answering with pretrained language models. The key idea is that instead of asking language models to directly generate answers to factoid questions, ask it to first generate a paragraph of text which will contain the answer ("recitation"). After recitation is complete, language models are instructed (via prompt examples) to use their recitation to answer the question. The paper additionally adopts several tricks to improve the performance of their system, including "self consistency" (generating through multiple recitations and taking majority vote), multiple recitations (for multi-hop questions). Finally, the authors explore fine-tuning a model to generate diverse recitations via passage hints. The authors use synthetic data generated by the frozen LM to achieve this goal.

The authors conduct experiments on Natural Questions, TriviaQA and HotpotQA, and find performance improvements over standard prompting techniques.

**Summary Of The Review:**

The paper has interesting ideas and surprising results, but I have two main concerns - (1) the paper does not evaluate the method on larger LMs which are available; (2) I don't think there's justification that this method is a replacement for retrieval in any way (weakness #2, #3).

I am currently leaning reject, but will be happy to move to the accept range if weakness #1 is addressed via experiments on GPT3-170B and InstructGPT3-170B.

------

**After rebuttal**: Thanks to the authors for the very detailed response! I've decided to raise my score to 6 (accept range) due to the improvements shown on Codex. I would still suggest the authors to take a more balanced take in their conclusion, mentioning that while there are improvements over direct generation, there is still a gap behind retrieval-augmentation on NQ.

---

> ### Author Response · Authors · 2022-11-18
> **Response to Reviewer HxiC**
>
> We thank the reviewer for their time, insightful comments, and questions. We have provided our responses below.
>
> > The paper would be much stronger with experiments on GPT3, Instruct-GPT3 (davinci-002), and larger language models (larger in-house LMs?). It's not really clear from the paper whether recitation helps with larger scale, which I think is important for the generalizability of the method [1]. This could work both ways --- I'm suspecting larger LMs will be better at both recitation and directly performing QA. I think experiments on InstructGPT [4], T0 [3] or FLAN [2, 7] will be especially interesting, since it's been fine-tuned on instructions / examples / human preferences.
>
> We presented new results on Codex, which is a public model with 175B parameters, where recite-and-answer consistenctly shows significant improvement on NQ and HotpotQA:
>
> | EM / F1  | Codex (direct) | Codex (chain-of-thought) | Codex (recite-and-answer) |
> |----------|----------------|--------------------------|---------------------------|
> | NQ       | 30.96 / 42.77  | -                        | **33.98 / 47.48**         |
> | TriviaQA | 83.40 / 87.12  | -                        | **83.50 / 87.97**         |
> | HotpotQA | 29.10 / 39.86  | 33.87 / 45.27            | **37.60 / 48.62**         |
>
> We also provide additional analysis of increasing model sizes (small, base, large, xl, xxl) on an instruction-finetuned language model (FLAN-T5) in Appendix F and Figure 6. We show that recite-and-answer prompting achieves consistent improvement over standard prompting, while the largest improvement is achieved in the largest “xxl (11B)” setting.
>
> > A major advantage of retrieval augmented systems is their applicability on (1) tail distribution information; (2) generalization to information which was not present in the model's training set (like COVID for BERT). I think these are important limitations of the proposed method, and (1) is not really discussed (2 is just mentioned in the conclusion). Are most of the correct recitations cases which were seen often during training?
>
> Thanks for the suggestion, we acknowledge that retrieval-augmented models do have their unique benefits in terms of tail information and generalization. However, as we mentioned in the introduction, retrieval-augmented models also have their own disadvantages, like requiring a known corpus to retrieve from, as well as a retrieval model. In this paper, we want to explore the possibility of reciting from the language model alone, without any retrieval modules. We think our exploration is a complementary direction to retrieval-augmented models, and maybe in the future, the two can be combined with each other to achieve stronger performance in the end.
>
> For most of our experiments, we use a pre-trained frozen model and perform in-context few-shot learning. So the recitation will not be seen during few-shot learning.
>
> > Overall, the performance of closed-book models in this paper seems to significantly lag behind recent few-shot retrieval-augmented systems [5, 6]...
>
> Please see our response above.
>
> > I have mixed thoughts about the passage hints fine-tuning experiments, since it requires fine-tuning a large LM on Wikipedia data. Perhaps the performance gains are because of the dedicated fine-tuning on Wikipedia data for the recitation LM model (which makes it overfit to Wikipedia)? Did you remove the passages from the test set questions while doing this fine-tuning? Also I don't think enough experiments are done in the paper to justify its added complexity over vanilla LM-Recitation. I would suggest moving it to the appendix, or performing experiments on all 3 datasets / models to show its benefit.
>
> Sorry for the confusion. The passage-hints fine-tuning experiments are not aimed for overfitting the corpus, rather we aim to show that in cases where there is a domain shift, e.g., we want to test the model on certain domains never seen in pre-training (medical or technical), then we can further fine-tune the model to incorporate an external corpus, such that the final model can recite from that corpus and perform well on those external domains. In some sense, we can think of this procedure as "embedding the external corpus into the LLM" so we have a unified model for recitation instead of doing "retrieve-and-answer" separately.
>
> > This paper is relevant to https://arxiv.org/abs/2004.05483 and https://arxiv.org/pdf/2110.08387.pdf, it would be great to cite them.
>
> We thank the reviewer for the suggestion. We have cited them in our paper.
>
> > In Table 4 (LM-Recitation_5), why is the number for different from Table 1 (Recite and answer)? (16.34 in Table 1 vs 14.16 in Table 4)
>
> We thank the reviewer for pointing it out. We accidentally report the NQ performance of UL2 with a later checkpoint (used for fine-tuning), while the rest results are reported with the Huggingface checkpoint. We have fixed Table 1 with the HF checkpoint result.

---

> > ### Comment · Reviewer_HxiC · 2022-11-18
> > **Thank you, raised score to 6**
> >
> > Thanks to the authors for the very detailed response! I've decided to raise my score to 6 (accept range) due to the improvements shown on Codex. I would still suggest the authors to take a more balanced take in their conclusion, mentioning that while there are improvements over direct generation, there is still a gap behind retrieval-augmentation on NQ.

---

### Official Review · Reviewer_rRR1 · 2022-10-24

**Confidence:** 4
**Clarity, Quality, Novelty And Reproducibility:** Mostly clear and novel.
**Correctness:** 4
**Technical Novelty And Significance:** 2
**Empirical Novelty And Significance:** 3
**Recommendation:** 6

**Strength And Weaknesses:**

(Strengths)

(1) Intuitive ideas

(2) No need to retrieve documents.

(3) Compatible with other approaches like chain-of-thought prompting.



(Weaknesses)

(1) Major concerns that will be listed below.

(2) Unclear cost for running the proposed methods against the standard-prompting models.

(3) Insufficient rationale to support gains and no gains

**Summary Of The Paper:**

The authors utilize LLM to first retrieve relevant information from LLM’s own memory (by sampling), then accomplishing more accurate factual generation to produce final outcomes. The core idea is motivated by the intuition: recite-step (retrieval) that recollects relevant knowledge pieces helps answer-step (generation) better output to knowledge intensive questions. Two models are provided:

For the prompt-based model,
(a) It first performs in-context learning for generating recitation provided with few-shot question-recitation examples.
(b) Then it performs another in-context learning to generate the right answer provided with the corresponding recitation and question as well as the question-answer pairs
(c) For a given question, the model samples multiple recitations (by (a)) and greedy-decode the answers (by (b)) for each sampled recitation.
(d) The model could do multi-hop reasoning by sequentially generating recitation (by simple number prompting) till the special tokens occurs, then finding the answer based on chain-reasoning.

For the passage-hint model,
(a) For each question-answer pair, prepare several Wikipedia pages that are ranked at the top (from a search engine) for the question query.
(b) Create the corresponding passage hint by concatenating hierarchical title orders from the topic and section/subsection titles to the position of the paragraph in the (sub)section.
(c) Given evidence-question pairs, perform in-context learning to sample a new question given an evidence passage.
(d) Learn a fine-tuned LM that outputs passage hint (from (b)) as well as evidence passage (from (c)) given a question input.
(e) Given a question, fine-tuned LM first samples multiple passage hints and passage evidence (by (d)).
(f) Aggregate multiple passage evidences (in (e)) as a diverse recitation to the given question, then generating the answer with a few-more question-answer demonstrations in a frozen language model.

The authors verify that the recite-and-answer method is effective for Closed-Book Question-Answering tasks being compatible with other useful approach that improve few-shot in-context learning capability.

**Summary Of The Review:**

(1) As the passage-hint model is a corpus-specific treatment that additionally utilizes title hierarchies, the prompt-based model must have great added benefits. The core concern is that the model performs in-context learning based on the model-generated recitations. No matter how you tune the temperature parameter, there must be a pathological trade-off between sacrificing diversity and sacrificing factuality. The current draft does not address impacts of less-accurate or incorrect recitations.

(2) It will be great to have the performance graph (given two metrics) over increasing number of few-shot prompts. If you could approximately measure the correctness of generated recitations, another figure that demonstrate the performance dependency with respect to the number of correct shots will benefit readers.

(3) Another major concern is the computational trade-off from using many shots. Table 1 shows that the proposed recite-and-answer model (with 5 shots) does not have significant impacts against standard-prompting (with more shots). Can you address the claim that using the standard-models with more shots is more expensive than running your prompt-based model with less-shots?

(4) It is less clear that how you decide the number of shots you would use for standard-prompting model and your recite-and-answer model. For example, UL2-20B is tested with 16 shots, whereas the in-house version used 64 shots. Justify the rationale of picking the number of shots and the reason not to test on more shots for OPT-30B.

(5) Excluding UL2-20B (mostly encoder-decoder T5 structure except the mixture of denoiser variations with multiple special tokens), is your In-house model of decoder-only? If so, why In-house has significant performance gain on HotpotQA but no visible boost on TriviaQA, which is the entirely opposite to OPT-30B? Any specific example-based explanation than a conjecture would be beneficial.

(6) The passage-hint model consists of significantly more steps. It will be useful to see overall time-cost trade-off between your model and standard models.

(7) Is the passage-hint model flexible if the evidence documents do not have a clear and fine-grained title/subtitle structures? What if some of title components from this hierarchy is missing? Should we expect performance degradation or performance boost because the passage hints are also generated being possibly incorrectly if the information structure is too granular.

(8) Overall, not much qualitative examples are provided. Having some successful and failure cases by examples will be highly

---

> ### Author Response · Authors · 2022-11-18
> **Response to Reviewer rRR1 (I of II)**
>
> We appreciate the in-depth questions and suggestions given by the reviewer. We have provided our responses below.
>
> > (1) ... No matter how you tune the temperature parameter, there must be a pathological trade-off between sacrificing diversity and sacrificing factuality. The current draft does not address impacts of less-accurate or incorrect recitations.
>
> We used a fixed 0.7 temperature for sampling for all experiments in the paper, following the recommended recipe in the self-consistency paper [1], which reports that with 20 reasoning paths, the temperature T and the top-k threshold k tend to be robust. For example, T = 0.3, 0.5, or 0.7 produce similar final results in terms of accuracy.
>
> > (2) It will be great to have the performance graph (given two metrics) over increasing number of few-shot prompts. If you could approximately measure the correctness of generated recitations, another figure that demonstrate the performance dependency with respect to the number of correct shots will benefit readers.
>
> We thank the reviewer for the insightful question. We provide additional analysis of the increasing number of few-shot prompts in Appendix E and Figure 5. We can see that recite-and-answer prompting achieves consistent improvement over standard prompting, while the largest improvement is achieved in the 1-shot setting.
>
> We also perform additional per-path analysis in Table 6. We can see that around $7\% \sim 10\%$ questions have the correct recitation but cannot produce the correct answer, while around $12\%$ questions do not have the correct recitation but can be answered correctly anyway.
>
> > (3) Can you address the claim that using the standard-models with more shots is more expensive than running your prompt-based model with less-shots?
>
> For few-shot in-context learning, the fair comparison would be between the same number of shots as existing work has shown that model performance is highly sensitive to the number of shots / the exact exemplars used.
>
> >  (4) It is less clear that how you decide the number of shots you would use for standard-prompting model and your recite-and-answer model.
>
> The number of shots is mainly decided by the maximum context length of different models. For example, the maximum sequence length of In-house LM is 2048, while the maximum sequence length of UL2 is 512. For OPT and Codex, we stick with the 5-shot setting to ensure a fair comparison across different models.
>
> > (5) Excluding UL2-20B (mostly encoder-decoder T5 structure except the mixture of denoiser variations with multiple special tokens), is your In-house model of decoder-only? If so, why In-house has significant performance gain on HotpotQA but no visible boost on TriviaQA, which is the entirely opposite to OPT-30B? Any specific example-based explanation than a conjecture would be beneficial.
>
> Yes, our in-house model is decoder-only. In our additional results on Codex, we can see that it has the same pattern as In-house LM-62B, that is, no visible boost on TriviaQA, and significant performance gain on HotpotQA. The reason behind this is that the TrviaQA questions are mostly in the common trivia question form, which may elicit certain QA patterns in pre-training that do not require explicit recitation, while HotpotQA questions are manually created for multi-hop reasoning, which requires the LM’s question decomposition ability, which is an emergent ability in large LMs. We additionally present two qualitative HotpotQA results on Codex in Figure 16 and Figure 17.
>
> > (6) The passage-hint model consists of significantly more steps. It will be useful to see overall time-cost trade-off between your model and standard models.
>
> Since the passage hint is usually short (e.g., concatenation of section titles), The passage-hint-based model would have similar decoding latency as prompt-based recite-and-answer models.
>
> > (7) Is the passage-hint model flexible if the evidence documents do not have a clear and fine-grained title/subtitle structures? What if some of title components from this hierarchy is missing? Should we expect performance degradation or performance boost because the passage hints are also generated being possibly incorrectly if the information structure is too granular.
>
> Note that our "passage-hint" framework is actually quite general -- in cases where documents don't have a clear structure, each paragraph can be encoded as "title -- #paragraph" where #paragraph can just be the index of each paragraph. Our model can benefit more when the structure is more fine-grained, i.e., it learns more information from subtitles/section titles if they exist.
>
> [1] Wang, Xuezhi, Jason Wei, Dale Schuurmans, Quoc Le, Ed Chi, and Denny Zhou. "Self-consistency improves chain of thought reasoning in language models." arXiv preprint arXiv:2203.11171 (2022).

---

> ### Author Response · Authors · 2022-11-18
> **Response to Reviewer rRR1 (II of II)**
>
> > (8) Overall, not much qualitative examples are provided. Having some successful and failure cases by examples will be highly
>
> We additionally present two qualitative HotpotQA results on Codex in Figure 16 and Figure 17. For example, here are two failure examples of direct answer and chain-of-thought and one successful example of recite-and-answer:
>
> ---
>
> Question: The director of the romantic comedy "Big Stone Gap" is based in what New York city?
>
> ---
>
> Answer (standard prompting): New York City
>
> ---
>
> Answer (chain-of-thought path-1): Adriana Trigiani is the director of the romantic comedy Big Stone Gap. Adriana Trigiani is based in New York City. The answer is New York City.
>
> ---
>
> Answer (recite-and-answer path-1):
>
> Answer 1: Big Stone Gap is a 2014 American romantic comedy film directed by Adriana Trigiani.
>
> Answer 2: Adriana Trigiani is an American author, playwright, filmmaker, and entrepreneur. She lives in
> Greenwich Village, New York City.
>
> The answer is Greenwich Village.
>
> ----
>
> We hope those examples can help the readers better understand how recite-and-answer helps LM generate more factually accurate answers.

---

### Official Review · Reviewer_9YM6 · 2022-10-25

**Confidence:** 3
**Correctness:** 3
**Technical Novelty And Significance:** 3
**Empirical Novelty And Significance:** Not applicable
**Recommendation:** 6

**Clarity, Quality, Novelty And Reproducibility:**

Both the core idea and writing are both clear and comprehensible. Missing details may well be compensated by open sourcing the code for reproducibility. The idea itself is well-motivated and novel.

**Strength And Weaknesses:**

Strengths
- The proposed method is intuitive and well motivated.
- Evaluation is sensible and yield positive results both wrt performance and data/resource efficiency.
- Writing is mostly clear.

Weaknesses
- The analysis section on what works and why is a bit thin; Table 5 overall is quite confusing. I believe answering the following questions  may help add more clarity and strength to this part.
  - What does Hits mean? What's the difference between @1 and @20? What is Hits@1 (no explanation in text)? What is (EM) following Hits@1? I'm guessing that's the subset of examples that the model answers correctly?
  - If I'm guessing correctly, the definition of Not Recit, hits@20-recit., and hits@20-path follows a binary decision tree, on whether the ground truth answer appears in any recitation or any QA output. It is rather difficult to figure this out based on the text in 4.3.4. Maybe a logical tree of some sort can help improve readability here.
  - Are there cases where, e.g., the ground-truth answer is not recited but the model still answers the question correctly? E.g., a more detailed characterization of Hits@1 would be interesting, unless reciting the gt answer is an empirical necessity – which by itself is interesting to point out if true.
  - How are the 1024 examples selected?
  - On a more minor note, the names for categories are very not self-explanatory and could be improved for better readability.
- There are some missing details that might slightly hurt readability
  - Judging from Fig 2-(1), <Recitation N> is generated from – and directly related to – <Question N>, with other Q-R pairs as in-context examples. Moving on to Fig 2-(2), it seems there are multiple QA pairs (Q0, A0, ...) inserted between <Recitation N> and <Question N>. Are the questions in these QA pairs the same as those in the QR pairs in the top-left block of Fig 2? Are they somehow related to <Question N>? If not, is there going to be some long-distance memory issues between <Recitation N> and <Question N>?
  - In Fig 2-(3), are the multiple paths (containing Recitations-dots-Answers) repeating Fig 2-(2)?
  - In the multi-hop setting, do the numbered prompts like "Recitation 1" and "Recitation 2" naturally give rise to generating recitations of different topics, or does one have to train/update the LM to solicit such behaviour?
  - "Since the multiple recited passages are generated in one-pass from the LLM decoding sequentially, ..." How to use multiple numbered prompts to generate multiple recitations in one-pass sequentially?
  - I'm personally curious about more details on the generated recitations, e.g., in NQ dataset, do the recitations resemble the "ground truth" passages in number of words/sentences/paragraphs? What percentage of recitations contain the correct answers?
- Minor points on motivations
  - One motivation mentioned for recitation is to match the form of causal LM pre-training objective (with the decimal of $\pi$ example). Personally, I tend to believe that recitation helps by making certain knowledge more explicit (value of $\pi$), and when primed on this explicit piece of information (in the form of recitation), the LM is then more likely to arrive at the correct answer (the 10th decimal of $\pi$). In other words, I'd argue that with or without recitation, the objective remains the same (i.e., masked or next-word prediction), so this motivation to me feels a bit mis-directed.
  - "Human's ability to recite relevant factoid knowledge before answering knowledge-intensive questions." Do humans have this ability? Regardless of the answer, having this ability doesn't necessarily help humans answer knowledge-intensive questions more accurately, which I guess should be the true human-inspired motivation for the proposal. In any case, citations are perhaps needed to make either claim.

**Summary Of The Paper:**

The paper proposed a LLM-based method to improve accuracy in factual knowledge generation. Instead of retrieving from external resources, the proposal trains a model to recite relevant knowledge given a query and subsequently answer the query conditioned on the recitation. The proposal is empirically shown to be effective on the closed book question answering task, with some ablation and analyses on various model settings.

**Summary Of The Review:**

The paper is of overall good quality, with a sound idea and good execution by putting many moving parts together into a working system with empirical gains. The analysis part is a bit unclear and underwhelming.

---

> ### Author Response · Authors · 2022-11-18
> **Response to Reviewer 9YM6**
>
> We thank the reviewer for their time, insightful comments, and questions. We have provided our responses below.
>
> > What does Hits mean? What's the difference between @1 and @20? What is Hits@1 (no explanation in text)? What is (EM) following Hits@1? I'm guessing that's the subset of examples that the model answers correctly?
>
> We have replaced the notation of Hits@1 (EM) with ``Hits@Majority (i.e., Exact Matching)’’, which means that the majority-voted answer is the same as the ground-truth annotation. An algorithmic description for error analysis is given in Algo. 1 in the appendix. We hope these clarifications are helpful.
>
> > Are there cases where, e.g., the ground-truth answer is not recited but the model still answers the question correctly? E.g., a more detailed characterization of Hits@1 would be interesting, unless reciting the gt answer is an empirical necessity – which by itself is interesting to point out if true.
>
> We thank the reviewer for the insightful question. We perform an additional per-path error analysis in Table 6. We can see that around 7%-10% of questions have the correct recitation but cannot produce the correct answer, while around 12% of questions do not have the correct recitation but can be answered correctly anyway.
>
> > How are the 1024 examples selected?
>
> Since most datasets are too large to run large language models on, especially with 20-path self-consistency evaluation, we follow [1] and use the top 1,024 data points for evaluation.
>
> > Are the questions in these QA pairs the same as those in the QR pairs in the top-left block of Fig 2? Are they somehow related to <Question N>? If not, is there going to be some long-distance memory issues between <Recitation N> and <Question N>?
>
> Yes, we use the same set of training questions as few-shot in-context examples for both recitation generation and recitation-augmented answer generation. They are not necessarily related to <Question N>. In practice, for all four language models, we do not see long-distance memory issues, i.e., recitation-augmented answer generation always produces more accurate results than direct answer generation.
>
> > In Fig 2-(3), are the multiple paths (containing Recitations-dots-Answers) repeating Fig 2-(2)?
>
> Yes. Self-consistency is sampling recitations and generating answers with different random seeds.
>
> > In the multi-hop setting, do the numbered prompts like "Recitation 1" and "Recitation 2" naturally give rise to generating recitations of different topics, or does one have to train/update the LM to solicit such behaviour?
>
> In the paper, we only report multi-hop results with few-shot in-context learning, so the LM is pre-trained and does not need to be updated. Empirically, "Recitation 1" and "Recitation 2" do tend to generate different (or complementary) topics, as shown in Figure 13.
>
> > "Since the multiple recited passages are generated in one-pass from the LLM decoding sequentially, ..." How to use multiple numbered prompts to generate multiple recitations in one-pass sequentially?
>
> Please check the 4-shot prompt we used for performing multiple-evidence recitation on the HotpotQA dataset in Figure 11.
>
> > I'm personally curious about more details on the generated recitations, e.g., in NQ dataset, do the recitations resemble the "ground truth" passages in number of words/sentences/paragraphs? What percentage of recitations contain the correct answers?
>
> We provide per-question and per-path error analysis in Table 5 and Table 6, respectively. When evaluating the UL2/OPT models on TriviaQA, in general, 40% of the recitations contain the correct answer, while for each question, ~80% of the questions can have the correct answer in one of their 20-path recitations at least once.
>
> > "Human's ability to recite relevant factoid knowledge before answering knowledge-intensive questions." Do humans have this ability? Regardless of the answer, having this ability doesn't necessarily help humans answer knowledge-intensive questions more accurately, which I guess should be the true human-inspired motivation for the proposal. In any case, citations are perhaps needed to make either claim.
>
> We thank the reviewer for the insightful suggestion. We have added a citation on how recitation (read-recite-review strategy) can help college students learn educational texts better [2].
>
> [1]: Wang, Xuezhi, Jason Wei, Dale Schuurmans, Quoc Le, Ed Chi, and Denny Zhou. "Rationale-Augmented Ensembles in Language Models." arXiv preprint arXiv:2207.00747 (2022).
>
> [2] McDaniel, Mark A., Daniel C. Howard, and Gilles O. Einstein. "The read-recite-review study strategy: Effective and portable." Psychological Science 20, no. 4 (2009): 516-522.

---

### Author Response · Authors · 2022-11-18
**General Response**

We thank all the reviewers for their precious time and insightful comments. We appreciate that the reviewers recognize our work as a well-motivated/intuitive/interesting idea (Reviewer 9YM6, Reviewer rRR1, Reviewer HxiC), written clearly (Reviewer 9YM6, Reviewer m3Mw), novel (Reviewer 9YM6), and having sensible/positive evaluation (Reviewer 9YM6, Reviewer HxiC). To improve the paper quality, we respond to the reviewers’ comments by making the following major revisions to the paper:

1. We presented new results on Codex-(a public model with 175B? parameters) in the paper, where recite-and-answer shows significant improvement on NQ and HotpotQA. This demonstrates the generalizability of the method.
2. We perform an additional per-path error analysis in Table 6. We can see that around 7%-10% of questions have the correct recitation but cannot produce the correct answer, while around 12% of questions do not have the correct recitation but can be answered correctly anyway. That means LM can still answer a portion of questions without the correct recitation.
3. We provide additional analysis of the increasing number of few-shot prompts in Appendix E and Figure 5. We can see that recite-and-answer prompting achieves consistent improvement over standard prompting, while the largest improvement is achieved in the 1-shot setting.
4. We provide additional analysis of increasing model sizes (small, base, large, xl, xxl) on an instruction-finetuned language model (FLAN-T5) in Appendix F and Figure 6. We show that recite-and-answer prompting achieves consistent improvement over standard prompting, while the largest improvement is achieved in the largest “xxl (11B)” setting.
5. We additionally present two qualitative HotpotQA results on Codex in Figure 16 and Figure 17. We additionally present two qualitative HotpotQA results on Codex in Figure 16 and Figure 17. For example, here are two failure examples of direct answer and chain-of-thought and one successful example of recite-and-answer:

---

Question: The director of the romantic comedy "Big Stone Gap" is based in what New York city?

---

Answer (standard prompting): New York City

---

Answer (chain-of-thought path-1): Adriana Trigiani is the director of the romantic comedy Big Stone Gap. Adriana Trigiani is based in New York City. The answer is New York City.

---

Answer (recite-and-answer path-1):

Answer 1: Big Stone Gap is a 2014 American romantic comedy film directed by Adriana Trigiani.

Answer 2: Adriana Trigiani is an American author, playwright, filmmaker, and entrepreneur. She lives in
Greenwich Village, New York City.

The answer is Greenwich Village.

----

We hope those examples can help the readers better understand how recite-and-answer helps LM generate more factually accurate answers.

---

### Decision · Program_Chairs · 2023-01-20

**Decision:**

Accept: poster

**Justification For Why Not Higher Score:**

If this paper is positioned as a scientific exploration, then more insights on why/how/when the proposed approach works or fails will be needed. If the paper is positioned as an alternative solution for the QA applications, then more practical considerations (e.g., efficiency, complexity, reliability, etc.) need to be evaluated and discussed in depth.

**Justification For Why Not Lower Score:**

It is a timely and trendy research topic. As the community is still learning and understanding the capabilities of LLMs, this paper is a nice addition in that it provides a uniquely different approach for open-domain QA in the closed-book setting.

**Metareview: Summary, Strengths And Weaknesses:**

Summary:
This paper proposes an interesting and novel approach for prompting Large Language Models (LLMs) for open-domain question answering. The key idea is to first ask the LLM to generate the support paragraphs that contain the answer (knowledge-recitation) and then use it as additional prompt, along with the question to ask the LLM to generate the answer (task-execution). Experiments are conducted on three QA datasets (Natural Questions, TriviaQA, and HotpotQA) in this closed-book setting, using three LMs  (In-house LM (62B), UL2 (20B), and OPT (30B)) and Codex (175B) in the updated version of the paper. The results shows better performance than direct prompting and chain-of-thoughts, and comparable performance to retrieval methods (BM25).

Strengths: All reviewers acknowledged that the paper is easy to follow. The proposed approach is a novel addition to the chain of thoughts line of work. The strong results in the closed-book QA setting is also somewhat surprising.

Weaknesses: While this work is definitely an interesting and novel scientific exploration of the capabilities of LLMs, the proposed approach is not a practical solution for open-domain QA. Some of the limitations are due to the closed book setting, such as outdated knowledge. Others are due to efficiency concerns and also the performance gap when compared to the open-book setting.


**Note From Pc:**

if the above contains the word "oral" or "spotlight" please see: "oral" presentation means -> notable-top-5% and "spotlight" means -> notable-top-25%. As stated in our emails, we are disassociating presentation type from AC recommendations